# Aorta smooth muscle-on-a-chip reveals impaired mitochondrial dynamics as a therapeutic target for aortic aneurysm in bicuspid aortic valve disease

Mieradilijiang Abudupataer[1], Shichao Zhu[1], Shiqiang Yan[2], Kehua Xu[2], Jingjing Zhang[2], Shaman Luo[2,3], Wenrui Ma[1], Md Fazle Alam[2,3], Yuyi Tang[2], Hui Huang[2], Nan Chen[1], Li Wang[2], Guoquan Yan[2], Jun Li[1,2], Hao Lai[1], Chunsheng Wang[1]\*, Kai Zhu[1]\*, Weijia Zhang[1,2,3]\*

[1]Department of Cardiac Surgery and Shanghai Institute of Cardiovascular Diseases, Zhongshan Hospital, Fudan University, Shanghai, China; [2]Institutes of Biomedical Sciences and the Shanghai Key Laboratory of Medical Epigenetics, Shanghai Medical College, Fudan University, Shanghai, China; [3]The State Key Laboratory of Molecular Engineering of Polymers, Fudan University, Shanghai, China

\*For correspondence:
wangchunsheng@fudan.edu.cn
(CW);
zhu.kai1@zs-hospital.sh.cn (KZ);
weijiazhang@fudan.edu.cn
(WZ)

Competing interests: The authors declare that no competing interests exist.

## Abstract

**Background:** Bicuspid aortic valve (BAV) is the most common congenital cardiovascular disease in general population and is frequently associated with the development of thoracic aortic aneurysm (TAA). There is no effective strategy to intervene with TAA progression due to an incomplete understanding of the pathogenesis. Insufficiency of NOTCH1 expression is highly related to BAV-TAA, but the underlying mechanism remains to be clarified.

**Methods:** A comparative proteomics analysis was used to explore the biological differences between non-diseased and BAV-TAA aortic tissues. A microfluidics-based aorta smooth muscle-on-a-chip model was constructed to evaluate the effect of NOTCH1 deficiency on contractile phenotype and mitochondrial dynamics of human aortic smooth muscle cells (HAoSMCs).

**Results:** Protein analyses of human aortic tissues showed the insufficient expression of NOTCH1 and impaired mitochondrial dynamics in BAV-TAA. HAoSMCs with NOTCH1-knockdown exhibited reduced contractile phenotype and were accompanied by attenuated mitochondrial fusion. Furthermore, we identified that mitochondrial fusion activators (leflunomide and teriflunomide) or mitochondrial fission inhibitor (Mdivi-1) partially rescued the disorders of mitochondrial dynamics in HAoSMCs derived from BAV-TAA patients.

**Conclusions:** The aorta smooth muscle-on-a-chip model simulates the human pathophysiological parameters of aorta biomechanics and provides a platform for molecular mechanism studies of aortic disease and related drug screening. This aorta smooth muscle-on-a-chip model and human tissue proteomic analysis revealed that impaired mitochondrial dynamics could be a potential therapeutic target for BAV-TAA.

**Funding:** National Key R and D Program of China, National Natural Science Foundation of China, Shanghai Municipal Science and Technology Major Project, Shanghai Science and Technology Commission, and Shanghai Municipal Education Commission.

## Introduction

Bicuspid aortic valve (BAV) disease is the most common congenital cardiovascular abnormality and is found in nearly 1.4% of the general population (*Garg et al., 2005*; *Michelena et al., 2011*;

**eLife digest** To function properly, the heart must remain a one-way system, pumping out oxygenated blood into the aorta – the largest artery in the body – so it can be distributed across the organism. The aortic valve, which sits at the entrance of the aorta, is a key component of this system. Its three flaps (or 'cusps') are pushed open when the blood exits the heart, and they shut tightly so it does not flow back in the incorrect direction.

Nearly 1.4% of people around the world are born with 'bicuspid' aortic valves that only have two flaps. These valves may harden or become leaky, forcing the heart to work harder. This defect is also associated with bulges on the aorta which progressively weaken the artery, sometimes causing it to rupture.

Open-heart surgery is currently the only way to treat these bulges (or 'aneurysms'), as no drug exists that could slow down disease progression. This is partly because the biological processes involved in the aneurysms worsening and bursting open is unclear.

Recent studies have highlighted that many individuals with bicuspid aortic valves also have lower levels of a protein known as NOTCH1, which plays a key signalling role for cells. Problems in the mitochondria – the structures that power up a cell – are also observed. However, it is not known how these findings are connected or linked with the aneurysms developing.

To answer this question, Abudupataer et al. analyzed the proteins present in diseased and healthy aortic muscle cells, confirming a lower production of NOTCH1 and impaired mitochondria in diseased tissues. They also created an 'aorta-on-a-chip' model where aortic muscle cells were grown in the laboratory under conditions resembling those found in the body – including the rhythmic strain that the aorta is under because of the heart beating. Abudupataer et al. then reduced NOTCH1 levels in healthy samples, which made the muscle tissue less able to contract and reduced the activity of the mitochondria. Applying drugs that tweak mitochondrial activity helped tissues from patients with bicuspid aortic valves to work better.

These compounds could potentially benefit individuals with deficient aortic valves, but experiments in animals and clinical trials would be needed first to confirm the results and assess safety. The aorta-on-a-chip model developed by Abudupataer et al. also provides a platform to screen for drugs and examine the molecular mechanisms at play in aortic diseases.

Verma and Siu, 2014). BAV arises from incomplete separation or fusion of the aortic valve cusps and is associated with an approximately 40% risk of developing thoracic aortic aneurysm (TAA), namely, bicuspid aortopathy (Verma and Siu, 2014). BAV-TAA poses a severe health threat to a large population because progressive aneurysmal dilation can potentially develop into lethal dissection or rupture (Goldfinger et al., 2014; Olsson et al., 2006). The current clinical management mainly relies on prophylactic surgical repair of the notably dilated aorta (Coady et al., 2010). At present, the understanding of pathophysiological mechanisms of BAV-TAA is incomplete, which leads to the absence of effective pharmaceutical therapy to alleviate aortopathy progression (Lindeman and Matsumura, 2019). Multiple factors, such as genetics and hemodynamics, are involved in the etiologies of BAV-TAA. In particular, genetic factors are considered to play a pivotal role in the disease progression (Isselbacher et al., 2016; Prakash et al., 2014). NOTCH1 insufficiency has been observed in the population with BAV (Balistreri et al., 2018; Harrison et al., 2019; Malashicheva et al., 2020; Sciacca et al., 2013). However, the underlying mechanism through which insufficient NOTCH1 induces aortopathy remains to be explored.

Mitochondrial dysfunction has been closely linked to a variety of cardiovascular disorders, such as heart failure and atherosclerosis. Recent studies found that mitochondrial dysfunction was also related to the development of arterial aneurysm formation (Cooper et al., 2021; van der Pluijm et al., 2018; Oller et al., 2021). A single-cell transcriptome analysis on aneurysmal human aortic tissue suggested that mitochondrial dysfunction and increased chromatin oxidative phosphorylation (OXPHOS) were found in TAA tissues and insufficient ATP production might not be sufficient for the contractile activities of human aortic smooth muscle cells (HAoSMCs) (Li et al., 2020). Particularly, mitochondrial fission and fusion are dynamically balanced to maintain mitochondrial homeostasis and functions; and a shift toward fission event is one of the main causes of mitochondrial

dysfunction. In a mice model of abdominal aortic aneurysm (AAA), impaired mitochondrial dynamics was found to play salient roles in disease development, and could be attenuated by the mitochondrial fission inhibitor Mdivi1 (*Cooper et al., 2021*). However, these studies focused on the analysis of abdominal aortic aneurysms and genetic TAA with FBN1 or Fubulin-4 mutation. It has been reported that there was close interaction between NOTCH1 signaling and homeostatic mitochondrial dynamics in the differentiation of cardiomyocytes (*Kasahara et al., 2013*) and the survival of breast cancer cells (*Chen et al., 2018*). Therefore, the relationship between NOTCH1 signaling pathway and mitochondrial dynamics in BAV-TAA needs to be clarified.

The traditional TAA animal models are frequently applied for the pathogenesis research and pharmaceutical therapy, however, not suitable for studying BAV-TAA. Although Koenig et al. generated NOTCH1-haploinsufficient mice in a preliminary 129S6 background that exhibited aortic root dilation, these mice did not show BAV characteristics (*Koenig et al., 2017*). Therefore, these models may not provide sufficient information of pathogenesis and drug response of BAV-TAA. Owing to the bioengineering advance, microfluidic-based organ-on-chip models have been widely developed to replicates human tissue microenvironment for toxicity analysis, drug screening and disease modeling and thus promotes pharmaceutical translation from preclinical studies to clinical trials (*Zhang et al., 2018*; *Ingber, 2016*; *Park et al., 2019*; *Thacker et al., 2020*; *Hofemeier et al., 2021*). It provides an opportunity to use a novel platform to study BAV-TAA on a susceptible human genetic background and may fill the gap between animal and human medicine. Here, we engineered an in vitro aorta smooth muscle-on-a-chip model of primary HAoSMCs that emulates the biomechanics of the human aortic wall. We characterized the association between mitochondrial dynamics and NOTCH1 deficiency in BAV-TAA on this platform. Our study provides the first demonstration previously undocumented role of impairment of mitochondrial fusion in bicuspid aortopathy, which may serve as a potential pharmacological target for preventing disease progression.

## Materials and methods

**Key resources table**

| Reagent type (species) or resource | Designation | Source or reference | Identifiers | Additional information |
|---|---|---|---|---|
| Gene (*Homo sapiens*) | NOTCH1 | NCBI | ID: 4851 | |
| Cell line (*Homo sapiens*) | Human aortic smooth muscle cell line | ATCC | CRL1999, Lot Number: 70019189, RRID:CVCL_4009 | Female, 11 months old, Caucasian |
| Cell line (*Homo sapiens*) | Primary human aortic smooth muscle cells | ATCC | PCS100012 | Male, 29 years old, African American |
| Cell line (*Homo sapiens*) | Primary human aortic smooth muscle cells | This paper | | Primary human aortic smooth muscle cells isolated from non-diseased and BAV-TAA patients (Asian). |
| Transfected construct (human) | NOTCH1 targeted shRNA | GeneChem (*Shao et al., 2015*). | | |
| Biological sample (human) | Ascending aorta | Zhongshan Hospital, Fudan University | | Ascending aortic tissues from non-diseased and BAV-TAA patients. |
| Antibody | Anti-DRP1 (D6C7) (Rabbit monoclonal) | Cell Signaling Technology | Cat # 8570S, RRID:AB_10950498 | (1:1000), western blotting |
| Antibody | Anti-MFF (E5W4M) (Rabbit monoclonal) | Cell Signaling Technology | Cat # 84580, RRID:AB_2728769 | (1:1000), western blotting |
| Antibody | Anti-Mitofusin-1 (D6E2S) (Rabbit monoclonal) | Cell Signaling Technology | Cat # 14739S, RRID:AB_2744531 | (1:1000), western blotting |

*Continued on next page*

*Continued*

| Reagent type (species) or resource | Designation | Source or reference | Identifiers | Additional information |
|---|---|---|---|---|
| Antibody | Anti-Mitofusin-2 (D1E9) (Rabbit monoclonal) | Cell Signaling Technology | Cat # 11925, RRID:AB_2750893 | (1:1000), western blotting |
| Antibody | Anti- Notch1 [EP1238Y] (Rabbit monoclonal) | Abcam | Cat # ab52627, RRID:AB_881725 | (1:1000), western blotting |
| Antibody | Anti- TAGLN (SM22) (Rabbit polyclonal) | Abcam | Cat # ab14106, AB_443021 | (1:1000), western blotting, (1:300) for IF |
| Antibody | Anti- Calponin (CNN1) (Rabbit monoclonal) | Abcam | Cat # ab46794, RRID:AB_2291941 | (1:1000), western blotting, (1:300) for IF |
| Commercial assay or kit | Tetramethylrhodamine methyl ester (TMRM) | Thermo Fisher Scientific | Cat # I34361 | |
| Commercial assay or kit | MitoSOX | Thermo Fisher Scientific | Cat # M36008 | |
| Commercial assay or kit | MitoTracker | Thermo Fisher Scientific | Cat # M22426 | |
| Chemical compound, drug | Mdivi-1 | Sigma | Cat # M0199 | a final concentration of 30 µM. |
| Chemical compound, drug | Leflunomide | Sigma | Cat # L5025 | a final concentration of 75 µM. |
| Chemical compound, drug | Teriflunomide | Sigma | Cat # SML0936 | a final concentration of 75 µM. |
| Other | Commercial flexible PDMS membrane | Hangzhou Bald Advanced Materials | # KYQ-200 | |

## Aorta smooth muscle-on-a-chip model design and fabrication

The structure of the three-layer microfluidic aorta smooth muscle-on-a-chip model was designed using computer-aided design (CAD) software (Autodesk Inc). The size of the three layers were 100 mm × 40 mm × 6 mm. The top and bottom layers had microchannels with dimensions of 70 mm × 6 mm × 4 mm, and the middle layer contained a microchannel with dimensions of 70 mm × 6 mm × 6 mm. Molds of the three layers were custom-made using a high-precision computer numerical control (CNC) engraving machine (Jingyan Technology). The frame of the molds and the microchannels were carved out of polymethyl methacrylate (PMMA) plates, which were then glued on another PMMA plate. Polydimethylsiloxane (PDMS, Sylgard 184, Dow Corning) was polymerized in defined casts at a weight ratio of base to curing agent of 10:1. The mixed PDMS was poured into the molds and underwent cross-linking at 70°C for 2 hr. Commercialized PDMS membranes were purchased from Hangzhou Bald Advanced Materials. Detailed parameters of the commercialized PDMS membrane were provided as follows: thickness of 200 ± 2 µm, shore A hardness of 50, Yang's elastic modulus of 1.7 MPa, tensile strength of 4 MPa, tear strength of 7 KN/m and light transmittance of 93%. The Young's modulus values were obtained by the following experiments. The tensile stress–strain responses was measured using a tensile testing machine (Instron). Prior to the measurement, the PDMS membrane was cut into a piece of 3 cm length and 3 mm width membrane. The membrane was fixed to the testing machine with a fixture. The sample was automatically stretched in a gradient within the proportional limit. The Young's modulus of the PDMS membrane was calculated by the slope value of the tensile stress–strain curve. Subsequently, the three PDMS layers were peeled off the molds. The bottom layer of the PDMS slab was bonded with one PDMS membrane after oxygen plasma treatment (Harrick Plasma), and the top PDMS layer was bonded to another PDMS membrane in a similar manner. The middle PDMS layer was then sandwiched between the

top and bottom membranes; this step was performed under a microscope to guarantee that the upper and lower microchannels fully overlapped with the middle microchannel.

## Mechanical stimulation

The cells were stretched by applying different percentages of rhythmic strain to the PDMS membranes for 24 hr. The vacuum pump was connected to a water-oil separator, which dried the gas to protect the downstream vacuum regulator and solenoid valve. The inlet of the vacuum pumps was then connected to the computer-controlled solenoid system by applying rhythmic stretching at a frequency of 1 Hz and then connected to the gas channel of the aorta smooth muscle-on-a-chip model. The regulator was used to control the vacuum magnitude. The solenoid valve was a voltage-dependent on/off valve used to control the stretching frequency of the PDMS membrane. When the supply voltage was greater than 24 V, the gas in the channel was pumped out, which stretched the PDMS membrane. Otherwise, the gas channel was connected to the atmosphere, and the membrane deformation recovered. The on/off frequency of the solenoid valve was controlled by a microcontroller unit (MCU). Thus, the stretching frequency could be controlled by changing the preset program of the MCU, and the stretching amplitude could be controlled by adjusting the vacuum regulator manually. We used two pressure ranges, 10 kPa (7.18 ± 0.44%, low strain) and 15 kPa (17.28 ± 0.91%, high strain), throughout the experiments. As the control, cells were cultured in aorta smooth muscle-on-a-chip models under static conditions. After 24 hr of rhythmic strain, samples were collected for immunofluorescence, RT-qPCR, western blotting and mitochondrial membrane dynamics analyses.

## Human aortic sample collection

Written informed consent was obtained from all patients before participation. Human aortic specimens were utilized under approvals of Zhongshan Hospital, Fudan University Ethics Committee (NO. B2020-158) in accordance with the Declaration of Helsinki. Human aortic samples were collected from patients who underwent ascending aorta surgery at Zhongshan Hospital, Fudan University. Echocardiography was used to characterize aortic valve morphology and ascending aortic diameter prior to surgery. The tissue samples were immediately frozen in liquid nitrogen and stored at −80℃. Six aortic tissues were obtained from patients with a tricuspid aortic valve but without aortic dilation (non-diseased; mean age: 62.2 years; range: 51–74 years; four males), and another six samples were obtained from patients with BAV-related thoracic aortic aneurysm (BAV-TAA; mean age: 59.3 years; range: 43–72 years; four males). The patients' basic information is available in *Supplementary file 1a*.

## Primary human smooth muscle cell isolation

Primary human aortic smooth muscle cells (p-HAoSMCs) were isolated from non-diseased ascending aortic tissues and BAV-TAA aortic tissues (n = 3). The ascending aortic tissues were washed with phosphate-buffered saline (PBS, Thermo Fisher Scientific). The intima and adventitia layers of the tissues were removed, and the media layer was preserved for the harvesting of p-HASMCs. Subsequently, the media layer was cut into small pieces (2–3 mm in length) and cultured in high-glucose Dulbecco's modified Eagle's medium (DMEM, Gibco) with 20% fetal bovine serum (FBS, Gibco) and 1% penicillin and streptomycin (p/s, Gibco) for 2–3 weeks at 37℃ and 5% $CO_2$ in a humidified incubator. After approximately 10–12 days, the p-HASMCs started to migrate out of the tissue pieces. When the cells reached approximately 80% confluency, first-passaged cells were rinsed with PBS, digested using 0.25% trypsin (Gibco), and replated in smooth muscle cell culture medium (SMCM, ScienCell). The cells were characterized through an immunofluorescence analysis of four different specific markers of smooth muscle cells (CNN1, SM22, MYH11 and α-SMA). We used p-HASMCs at a low passage (P2-P5) in all the experiments.

## Cell culture

In addition to the p-HAoSMCs isolated from aortic tissues, a human aortic smooth muscle cell line (CRL1999) and commercialized p-HAoSMCs (PCS-100–012) were purchased from ATCC (American Type Culture Collection) in accordance with their ethical regulations and compliances. p-HAoSMCs and the CRL1999 cell line were cultured in SMCM. Prior to cell seeding, the surface of the cell

culture channel was coated with mouse collagen at a concentration of 80 µg/mL (Sigma) by incubating for 1 hr at room temperature and drying for 2 hr at 70°C. Afterward, the cell culture channel was washed with PBS, and cells were seeded on the PDMS membranes in a cell culture channel at a density of 2 x 10$^6$ cells/mL. The cells were cultured in Dulbecco's modified Eagle's medium/nutrient mixture F-12 (DMEM/F-12, Thermo Fisher Scientific) supplemented with 10% FBS in a cell culture channel. After seeding, aorta smooth muscle-on-a-chip models were incubated at 37°C and 5% CO$_2$ in a humidified incubator for 24 hr for cell attachment. The aorta smooth muscle-on-a-chip models were then ready for mechanical stimulation experiments.

## Cell line

HAoSMCs cell line (CRL1999, Lot number 70019189, Homo sapiens) and primary HAoSMCs (PCS-100–012, Lot number 80323179, *Homo sapiens*) were purchased from ATCC. The identity has been authenticated by STR analysis and mycoplasma contamination was conformed by sterility test and pathogenic virus test provided by ATCC. The cells have human unique DNA profiles and were negative for mycoplasma contamination.

## Drug screening

Mdivi-1 (Sigma), an inhibitor of mitochondrial fission, was dissolved in dimethylsulfoxide (DMSO) and stored at −20°C before use, and fresh medium was used to obtain a final concentration of 30 µM. Leflunomide and teriflunomide (Sigma), two different activators of mitochondrial fusion, were dissolved in DMSO at appropriate concentrations. Prior to cell treatment, fresh medium was used to obtain a concentration of 75 µM for both drugs. After the cells were fully attached to the PDMS membrane in the cell culture channel, medium containing a mitochondrial fusion activator (leflunomide and teriflunomide) or mitochondrial fission inhibitor (Mdivi-1) was added to the aorta smooth muscle-on-a-chip models. After 24 hr of rhythmic strain, the samples were collected for comparative experiments.

## Immunofluorescence analysis

Immunofluorescence analysis was performed in the microfluidic aorta smooth muscle-on-a-chip model in situ after 24 hr of rhythmic stretch. The medium was aspirated from the cell culture channel, and the cells were washed with PBS, immediately fixed with 4% paraformaldehyde (Beyotime) for 30 min at room temperature and permeabilized with 1% (v/v) Triton X-100 (Beyotime) for 15 min. Afterward, blocking solution with 5% bovine serum albumin (Sigma) was applied to the cells to block nonspecific binding sites for 30 min at room temperature, and the cells were incubated overnight at 4°C with primary antibodies. The primary antibodies used in this study and their working concentrations are listed in *Supplementary file 1b*. After incubation, the cells were washed three times with PBS and incubated with Alexa 594 anti-rabbit secondary antibody (Thermo Fisher Scientific) at a dilution of 1:300 for 1 hr at room temperature under dark conditions. The nuclei were counterstained with 4′,6-diamidino-2-phenyllindole (DAPI) (Thermo Fisher Scientific) for 10 min. The aorta smooth muscle-on-a-chip models were then disassembled, and images were acquired with a fluorescence microscope (Leica DMi8) and analyzed using ImageJ software.

## Western blotting

Cells and aortic tissue samples were lysed using RIPA (Beyotime) lysis buffer supplemented with protease inhibitor phenyl methyl sulfonyl fluoride (PMSF, Beyotime). To collect an appropriate concentration of protein for western blotting, we collected and pooled cell protein from three microfluidic aorta smooth muscle-on-a-chip model. For the tissue samples, the intima and adventitia were peeled out, and the middle layers were ground into small pieces. The extracts were incubated for 30 min on ice for complete lysis and centrifuged at 14,000 rpm and 4°C for 25 min. The supernatant was collected after centrifugation, and the debris was discarded. The protein concentrations were quantified using a BCA Protein Assay kit (Thermo Fisher Scientific). The extracted proteins were diluted in sample loading buffer and heated for 5 min at 95°C. Ten micrograms of each protein sample were then separated by running on a 10% SDS-PAGE gel and subsequently transferred to 0.2 µm polyvinylidene fluoride (PVDF) membranes (Millipore). The PVDF membranes were blocked with 5% skimmed milk (Beyotime) for 1 hr at room temperature and incubated with different primary

antibodies (*Supplementary file 1b*) overnight at 4°C. The membranes were then incubated with horseradish peroxidase-conjugated goat anti-rabbit and goat anti-mouse IgG secondary antibodies (Cell Signaling Technology) at 1:6000 dilution for 1 hr at room temperature. Bands were detected using the SuperSignal chemiluminescence reagent substrate (Millipore). The protein expression level was normalized using the housekeeping protein β-actin. Quantitative estimation of the band intensity was performed using Image J software.

## Knockdown of NOTCH1 by lentivirus short hairpin RNA

NOTCH 1-targeted short hairpin RNA (shRNA) was designed and synthesized by GeneChem. p-HASMCs and CRL-1999 cells were seeded in six-well plates (6 x $10^5$ cells/well) and cultured in a humidified incubator at 37°C with 5% $CO_2$. When the cells reached 30% confluency, the cells were divided into three groups: control (WT), negative control shRNA (NC) and NOTCH1 knockdown (NOTCH1-KD). The cells in the NC and NOTCH1-KD groups were infected with lentivirus-nonspecific shRNA and lentivirus-shRNA-NOTCH1 at a multiplicity of infection (MOI) of 10 according to the manufacturer's recommended protocol (GeneChem). After 8–12 hr of infection, the virus particles were removed from the respective wells, and fresh SMCM was added. The cells were further cultured for 72 hr in a humidified incubator at 37°C with 5% $CO_2$. To establish a stable cell line, the cells were treated with puromycin at a concentration of 2 μg/mL for 5 days. When the cells reached approximately 80% confluency, the cells were harvested, and the knockdown efficiency of NOTCH1 was evaluated by qRT-PCR and western blotting assays.

## Morphology and membrane potential analyses of mitochondria

For analyses of the mitochondrial morphology and membrane potential, we used different types of fluorescent dyes, including tetramethylrhodamine methyl ester (TMRM), MitoSOX, and MitoTracker (all from Thermo Fisher Scientific), according to the manufacturer's guidelines. Cells were stretched on aorta smooth muscle-on-a-chip models for 24 hr. An appropriate concentration of each fluorescent dye was added to three different aorta smooth muscle-on-a-chip models and incubated at 37°C for 30 min in the dark. After incubation, the channels were washed three times with PBS, and the nuclei were counterstained with Hoechst (Sigma) for 10 min. Representative staining images of all three fluorescent dyes (under static and strain conditions) were acquired using a fluorescence microscope (Leica) and analyzed using Image J software.

## Estimation of the ATP concentration

The level of ATP production by cells on the aorta smooth muscle-on-a-chip models was determined using an ATP assay kit (Beyotime) following the manufacturer's instructions. In brief, after mechanical stimulation of the cells on the aorta smooth muscle-on-a-chip models, the cells were lysed using ATP lysis buffer, and total protein was collected by centrifugation of the cell lysate at 12,000 rpm and 4°C for 5 min. After centrifugation, the supernatant was collected, mixed with ATP detection reagent and incubated for 10 min at room temperature. After incubation, the ATP concentration was measured using a luminometer. An ATP concentration standard curve was then established and used to calculate the ATP concentration of each sample.

### Mass spectrometry analysis

The aortic samples were minced and lysed with RIPA on ice for 30 min. The extracts were centrifuged at 14,000 rpm and 4°C for 25 min, and the supernatant was collected after centrifugation. The protein concentrations were quantified using a BCA Protein Assay kit (Thermo Fisher Scientific). Filter-aided sample preparation (FASP) was performed for protein digestion. Before alkylation with 10 mM dithiothreitol (DTT, Sigma) and 30 mM iodoacetamide (IAA, Sigma), the proteins were loaded in 10 kDa centrifugal filter tubes (Millipore) and treated twice with 50 mM $NH_4HCO_3$ (Sigma). The extracts were digested with trypsin at a ratio of 1:50 and incubated at 37°C overnight. Trifluoroacetic acid (TFA, 0.1%) was added to stop the digestion reaction. One hundred micrograms of peptides containing 0.1% TFA was loaded in high-pH reversed-phase fractionation spin columns (Thermo Fisher Scientific). We obtained 10 flow-through fractions, and two fractions were combined to obtain one sample. The resulting five fractions were dried by vacuum centrifugation. The samples were resuspended in 30 μL of solvent A (A: water with 0.1% formic acid; B: ACN with 0.1% formic acid),

separated by nanoLC and analyzed by on-line electrospray tandem mass spectrometry. The experiments were performed using a nanoAquity UPLC system (Waters Corporation) connected to a quadrupole-Orbitrap mass spectrometer (Q Exactive HF) (Thermo Fisher Scientific) equipped with an online nanoelectrospray ion source. Two microliters of peptide sample were loaded onto an analytical column (Acclaim PepMap C18, 75 µm x 25 cm) and subsequently separated with a linear gradient from 5% B to 30% B over 110 min. The column flow rate was maintained at 300 nL/min, and the column temperature was maintained at 45℃. An electrospray voltage of 2.2 kV versus the inlet of the mass spectrometer was used.

The Q Exactive HF mass spectrometer was operated in the data-dependent mode to switch automatically between MS and MS/MS acquisition. Survey full-scan MS spectra (m/z 350–1500) were acquired with a mass resolution of 60 K. The automatic gain control (AGC) was set to 3000000 with a maximum injection time of 50 ms. Fifteen sequential high-energy collisional dissociation (HCD) MS/MS scans with a resolution of 15.0 K were acquired with the Orbitrap. The intensity threshold was 50,000, and the maximum injection time was 80 ms. The AGC target was set to 100,000, and the isolation window was 1.6 m/z. Ions with charge states of 2+, 3+, and 4+ were fragmented with a normalized collision energy (NCE) of 30%. In all cases, one microscan was recorded using dynamic exclusion of 20 s. In the MS/MS, the fixed first mass was set to 110. Online peaks were used for the analysis of proteomic data. The precursor mass error tolerance was set to 10 ppm with a fragment mass error tolerance of 0.05 Da. In all software programs, carbamidomethylation was set as a fixed modification, and variable modifications of oxidation (M), acetylation (N-term) and deamidation (NQ) were included. The false discovery rate (FDR) for peptide and protein identifications was set to 1%. Total ion chromatography (TIC) was used for normalization. The rest of the parameters were set to the default values. The MS/MS spectra were searched using the Andromeda search engine against the Swiss-Prot database (Taxonomy: *Homo sapiens*, Release 2020-11-02) (total of 20385 entries). The statistical analyses were mainly conducted in R 3.6.1. Label-free quantification (LFQ) was used for the following analysis flow. Proteins containing more than 50% missing values were removed, and the remaining missing values were inputted by k-nearest neighbor (kNN) imputation based on the Euclidean distance using the DMwR package in R. After normalizing the trimmed mean of M-values (TMM), PCA showed no significant batch effect. Differential expression analysis was conducted using the Limma package. The proteins with a p-value threshold of 0.05 and fold change > 1.5 were identified as differentially expressed proteins and inputted into the IPA. The mass spectrometry proteomics data have been deposited to the ProteomeXchange Consortium via the PRIDE partner repository with the dataset identifier PXD026303.

## Statistical analyses

The experimental results are expressed as the means ± standard deviations (SDs). A minimum of three individual replications of each group were used for the relative analyses. The statistical analyses were performed using GraphPad Prism eight software. Two-tailed Student's t tests were used to compare values between two groups, and one-way or two-way analysis of variance (ANOVA) followed by Tukey's post hoc test was used for multiple-group comparisons. Statistical significance was indicated by *p < 0.05, **p < 0.01, ***p < 0.001, and ****p < 0.0001.

## Results

### The relationship between NOTCH1 insufficiency and mitochondrial dysfunction in human TAA

To explore the pathological process involved in BAV-TAA, aortic tissues were collected from six patients with BAV-TAA who underwent ascending aorta replacement and six patients with non-diseased aorta who underwent cardiac surgery. The clinical characteristics of the patients are shown in *Supplementary file 1a*. Hematoxylin and eosin (H and E) staining showed interrupted elastic fibers and thinning of the tunica media in the BAV-TAA aortas compared with non-diseased aortic tissue (*Figure 1a*). A Western blotting analysis showed that NOTCH1 expression was significantly lower in BAV-TAA aortic tissues than in non-diseased aortic tissues. SM22 and CNN1 expression was significantly reduced in BAV-TAA aortic tissues (*Figure 1b–c*). We evaluated the expression of mitochondrial fission- and fusion-related proteins in non-diseased and BAV-TAA aortic tissues. The results

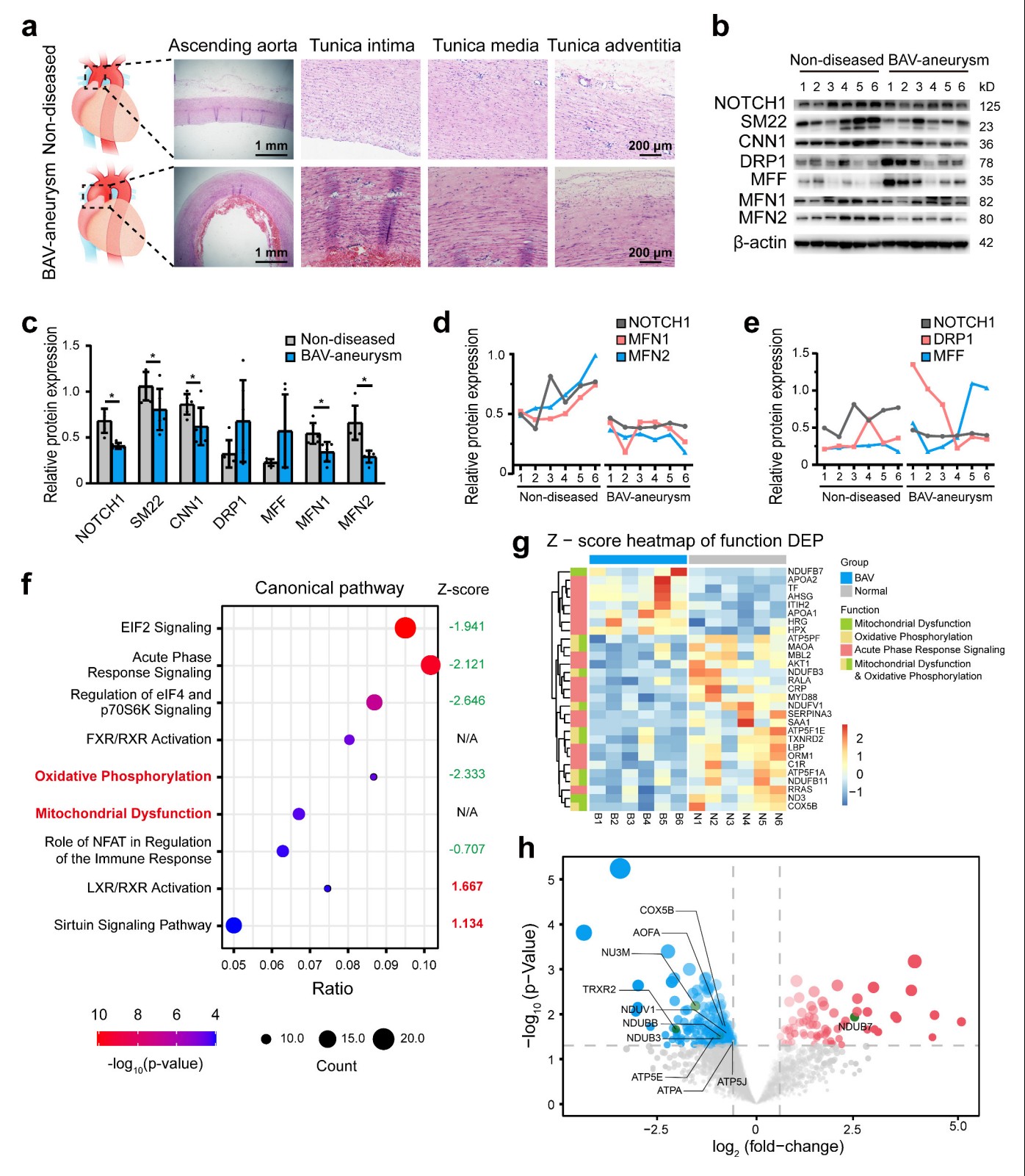

**Figure 1.** The relationship between NOTCH1 insufficiency and mitochondrial dysfunction in BAV-TAA. (a) H and E staining of non-diseased and BAV-TAA aortic tissues. The scale bar represents 1 mm in the low field and 200 μm in the high field. (b) Representative images of the western blotting analysis of the expression of NOTCH1, the mitochondrial dynamics proteins DRP-1, MFF, MFN1, and MFN2 and the contractile phenotype proteins SM22 and CNN1 in six non-diseased and six BAV-TAA aortic tissue fragments. (c) Quantification of the total band densities of the proteins normalized

*Figure 1 continued on next page*

*Figure 1 continued*

to the corresponding band density of β-actin (n = 6, *p < 0.05, two-tailed Student's t tests were used between two groups). (**d**) Correlation analysis among the quantified protein expression levels of NOTCH1, MFN1, and MFN2. A positive correlation was found between NOTCH1 and MFN1 or MFN2 in aortic tissues. The $R^2$ between NOTCH1 and MFN2 was 0.8069, and that between NOTCH1 and MFN1 was 0.6137. (**e**) Correlation analysis between the quantified protein expression of NOTCH1 and DRP1 or MFF. No correlation was found. (**f**) The enriched canonical pathways identified by IPA. (**g**) Heatmap of the expression of enriched proteins involved in mitochondrial dysfunction, the oxidative phosphorylation pathways and acute phase response signaling. (**h**) Volcano plot visualization of the differentially expressed proteins related to mitochondrial dysfunction. The colors indicate the following: gray, no differential expression; red, upregulated proteins; and blue, downregulated proteins. The proteins related to mitochondrial dysfunction are labeled. All the data are expressed as the means ± SDs.

The online version of this article includes the following source data for figure 1:

**Source data 1.** The original raw data for western blotting.
**Source data 2.** List of total differential protein expression between non-diseased and BAV-TAA aortic tissues.
**Source data 3.** The total enriched canonical pathways in BAV-TAA aortic tissues.
**Source data 4.** List of total protein expression between non-diseased and BAV-TAA aortic tissues.

showed that DRP1 and MFF expression was increased in the BAV-TAA group compared with the non-diseased group, but the differences were not significant probably due to individual patient differences, which resulted in relatively large protein expression differences within each group. However, the protein expression of MFN1 and MFN2 was significantly lower in the BAV-TAA group than in the non-diseased group (*Figure 1c*). In general, we found that MFN1 and MFN2 were expressed at low levels in tissues with NOTCH1 insufficiency (*Figure 1d*). The expression of NOTCH1 exhibited a positive correlation with MFN1 and MFN2 in aortic tissues. The expressions of DRP1 and MFF did not show a correlation with the expression of NOTCH1 (*Figure 1e*).

To further explore the biological differences between non-diseased and BAV-TAA aortic tissues, a comparative proteomics analysis of global proteins in aortic tissues was performed by high-performance liquid chromatography tandem mass spectrometry. In total, 70 upregulated proteins and 257 downregulated proteins were identified (*Supplementary file 2*). The enriched canonical pathways identified by Ingenuity Pathway Analysis (IPA) showed that acute phase response signaling, mitochondrial dysfunction and oxidative phosphorylation pathways were significantly enriched in BAV-TAA aortic tissues (*Figure 1f*, *Supplementary file 1c*). Among these pathways, Z-score of oxidative phosphorylation pathway was −2.333, indicating significantly inhibited. Z-score of mitochondria dysfunction was not applicable due to insufficient evidence in the knowledge base for confident activity predictions across datasets. In addition, metabolic signaling pathways affecting mitochondrial function, such as the EIF2 and sirtuin signaling pathways, were also significantly enriched in BAV-TAA. *Figure 1g* and *Supplementary file 1d* show the enriched proteins associated with mitochondrial dysfunction, oxidative phosphorylation pathways, and acute phase response signaling. In total, seven upregulated proteins and 11 downregulated proteins were found to be related to acute phase response signaling, and one upregulated protein and 10 downregulated proteins were associated with mitochondrial dysfunction (*Figure 1g*, *Supplementary file 1d*). *Figure 1h* shows a volcano plot of these 11 differentially expressed proteins that allows visualization of the fold change and p-value of all differentially expressed proteins between the two groups. MFN1, MFN2, and NOTCH1 were not detected by mass spectrometry analysis, mainly because the aortic tissues contain highly cross-linked extracellular matrix that can be refractory to protein extraction. The publicly available data of the most compressive clinical aortic proteome, up to now, also indicated the missingness of MFN1, MFN2, DRP1, and NOTCH1 protein in aortic specimens (*Herrington et al., 2018*).

## Construction of aorta smooth muscle-on-a-chip model

During cardiac systolic and diastolic cycles, the thoracic aortic wall experiences rhythmic tensile strain. Ascending aorta is the first section of the aorta, which starts from the left ventricle of the heart and extends to the aortic arch. It is connected to the left ventricular outflow track and is the part that pumps oxygenated blood to the body's tissues and organs. Clinical studies have shown that the circumferential strains of the aortic wall range from low values of 7.0 ± 2.5% to high values of 21.5 ± 12.4%, and these can be further influenced by age, the aortic diameter, and the presence of aortopathy (*Akazawa et al., 2016*; *Bell et al., 2014*). To better simulate the rhythmic tensile strain

experienced by HAoSMCs in vivo, we developed a compact microfluidic aorta smooth muscle-on-a-chip model with commercialized highly flexible polydimethylsiloxane (PDMS) membranes (**Figure 2a**). The model was composed of three chambers: (i) a top vacuum chamber deforming the upper PDMS membrane; (ii) a middle chamber containing the culture medium to maintain cell

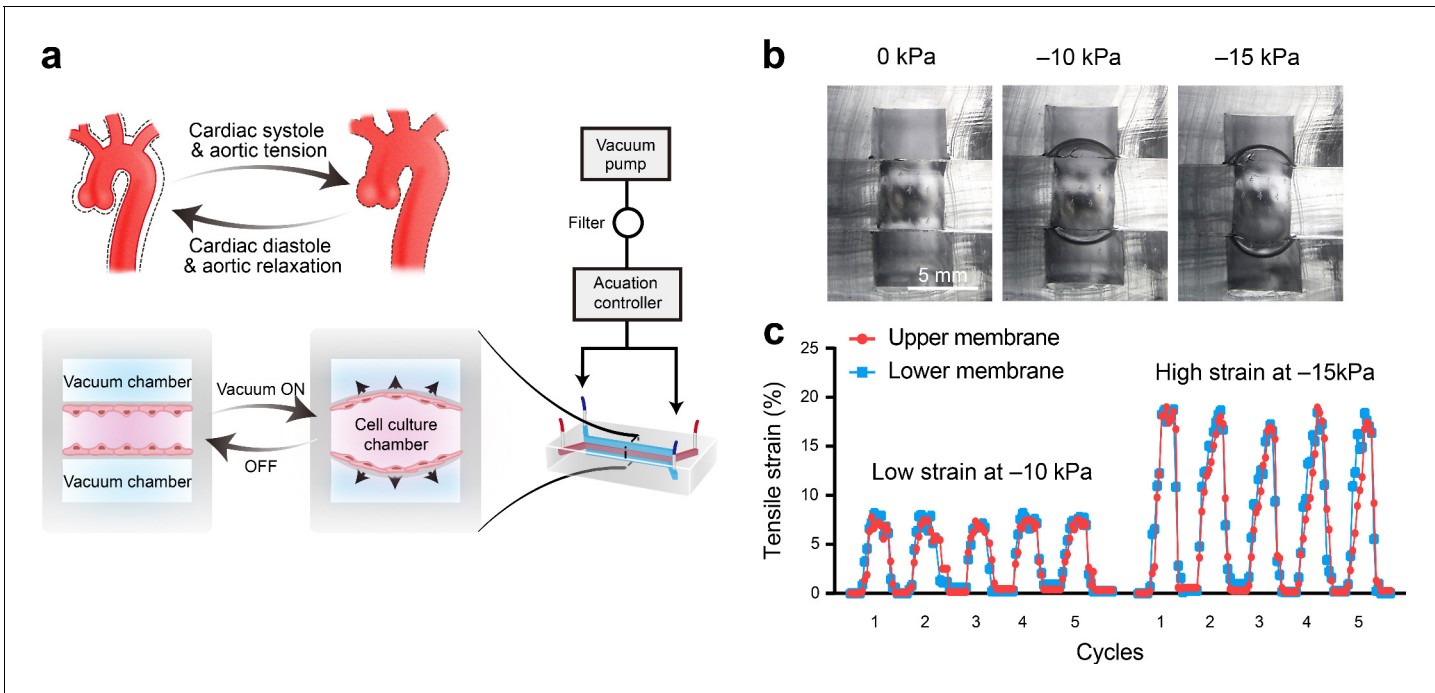

**Figure 2.** Schematic design and strain characterization of the aorta smooth muscle-on-a-chip model. (**a**) Schematic overview of the in vitro chip model. (**b**) Cross-sectional view of the microfluidic aorta smooth muscle-on-a-chip model showing the deformations of the PDMS membranes under different vacuum pressures. The scale bar represents 5 mm. (**c**) Measured tensile strains of the upper (red) and lower (blue) membranes at negative pressures of 10 kPa and 15 kPa for five cycles. The peak tensile strain per cycle averaged 7.18 ± 0.44% with a cyclic negative pressure of 10 kPa and 17.28 ± 0.91% with a cyclic negative pressure of 15 kPa.

The online version of this article includes the following source data and figure supplement(s) for figure 2:

**Source data 1.** The quantification data for measurement of tensile strains of the upper and lower membranes.

**Figure supplement 1.** Characterization and parameters of the aorta smooth muscle-on-a-chip model.

**Figure supplement 2.** Tensile stress–strain responses of PDMS membrane.

**Figure supplement 2—source data 1.** The raw data of Tensile stress–strain responses of PDMS membrane.

**Figure supplement 3.** The photographs of a piece of highly (500%) stretched PDMS membrane.

**Figure 2—animation 1.** Cross-sectional deformation of the PDMS membranes in the microfluidic chip with a 2 mm culturing channel, at the cyclic pressure of -10 kPa.

https://elifesciences.org/articles/69310#fig2video1

**Figure 2—animation 2.** Cross-sectional deformation of the PDMS membranes in the microfluidic chip with a 2 mm culturing channel, at the cyclic pressure of -15 kPa.

https://elifesciences.org/articles/69310#fig2video2

**Figure 2—animation 3.** Cross-sectional deformation of the PDMS membranes in the microfluidic chip with a 4 mm culturing channel, at the cyclic pressure of -10 kPa.

https://elifesciences.org/articles/69310#fig2video3

**Figure 2—animation 4.** Cross-sectional deformation of the PDMS membranes in the microfluidic chip with a 4 mm culturing channel, at the cyclic pressure of -15 kPa.

https://elifesciences.org/articles/69310#fig2video4

**Figure 2—animation 5.** Cross-sectional deformation of the PDMS membranes in the microfluidic chip with a 6 mm culturing channel, at the cyclic pressure of -10 kPa.

https://elifesciences.org/articles/69310#fig2video5

**Figure 2—animation 6.** Cross-sectional deformation of the PDMS membranes in the microfluidic chip with a 6 mm culturing channel, at the cyclic pressure of -15 kPa.

https://elifesciences.org/articles/69310#fig2video6

growth on the PDMS membranes; and (iii) a bottom vacuum chamber deforming the lower PDMS membrane. HAoSMCs were cultured on the PDMS membranes in the middle cell culture chamber. The dimensions of the model are shown in *Figure 2—figure supplement 1a*. The measured Young's elastic modulus values of the commercialized PDMS membrane were 1.71 MPa within 25% tensile strain and 1.67 MPa within 500% tensile strain (*Figure 2—figure supplement 2*) and it shows excellent homogeneity and tensile properties (*Figure 2—figure supplement 3*). The rhythmic tensile strain was generated by connecting the top and bottom chambers to a vacuum pump that cyclically deformed the PDMS membrane. The rhythm and value of the dynamic negative pressure in the chambers were controlled by a set of apparatuses consisting of a monochip computer, a pressure regulator and a solenoid valve.

To quantify the tensile strains of the PDMS membrane generated by negative pressure in a vacuum chamber, we captured the real-time deformations of the PDMS membranes and measured the changes in length. We captured the real-time deformations of the PDMS membranes from a cross-sectional view of the microfluidic model, with vacuum pressures of 0 kPa, 10 kPa, and 15 kPa, and measured the strain magnitude of the PDMS membrane (*Figure 2b*). The two deformations of the upper and lower PDMS membranes were coincident in terms of amplitudes and rhythms (*Figure 2c*). A vacuum pressure of 10 kPa induced 7.18 ± 0.44% strain (7.09 ± 0.18% strain in the lower layer and 7.27 ± 0.28% strain in the upper layer), and 15 kPa induced 17.28 ± 0.91% strain (17.29 ± 0.62% strain in the lower layer and 17.23 ± 0.64% strain in the upper layer) (*Figure 2c* and *Figure 2—figure supplement 1b*). We tested three prototypes with varying culturing channels, that is 2, or 4, or 6 mm in width (*Figure 2b–c* and *Figure 2—animations 1–6*). Finally, we opted for the largest size to harvest enough cells for protein analysis replication. To replicate different strains on the human aortic wall, we applied 7.18 ± 0.44% strain induced by a vacuum pressure of 10 kPa as a relatively low strain and 17.28 ± 0.91% strain induced by a vacuum pressure of 15 kPa as a relatively high strain.

## Rhythmic tensile strain defines cell alignment and enhances cell contractility

To identify the effect of rhythmic tensile strain on the contractility of HAoSMCs, the changes in cellular morphology, alignment and contractile/synthetic phenotypic markers were assessed under rhythmic low/high strain or static conditions (*Figure 3a*). Cytoskeletal F-actin staining images of HAoSMCs showed a decrease in the cell width and an increase in the cell length in the presence of rhythmic low or high strain (*Figure 3b*). The results also revealed an increase in the length-to-width ratio from 2.33 ± 0.82 under static conditions to 2.74 ± 1.01 under low strain or 3.50 ± 1.19 under high strain (*Figure 3c*). Compared with the irregular orientation of the cells observed under static culture, the cells tended to align perpendicularly to the direction of the applied strain. The angle between the directions of the cellular alignment and the applied strain was approximately 90° (*Figure 3d*). To evaluate the effect of rhythmic tensile strain on the expression of phenotypic markers, the protein levels of SM22, CNN1, and OPN in HAoSMCs were measured by western blotting. The results showed that the contractile phenotype markers SM22 and CNN1 were upregulated under either low or high strain (*Figure 3e–f*). The expression of the synthetic phenotypic marker OPN under both rhythmic low and high strain was lower than that observed under static conditions. These results indicate that the application of rhythmic strain can induce HAoSMCs to spread to longer shapes, align unidirectionally, and exhibit enhanced contractility on the chip model.

## Contractility and mitochondrial dynamics in NOTCH1-insufficient HAoSMCs

To clarify the effect of NOTCH1 insufficiency on HAoSMC contractility, we cultured NOTCH1-insufficient cells under rhythmic low or high strain or static conditions and then characterized the expression of phenotypic markers. The schematic workflow of the experimental design is shown in *Figure 4a*. First, NOTCH1 was knocked down in HAoSMCs using a lentivirus expressing short hairpin RNA (shRNA) targeting NOTCH1 (*Shao et al., 2015*). To evaluate the effect of the lentivirus vector on NOTCH1 and phenotypic markers, HAoSMCs without any treatments (WT), HAoSMCs treated with negative control shRNA (NC) and HAoSMCs treated with NOTCH1 shRNA (NOTCH1-KD) were verified using quantitative real-time PCR and western blotting experiments. An approximately 60% reduction in NOTCH1 mRNA expression was found in the NOTCH1-KD group compared with the

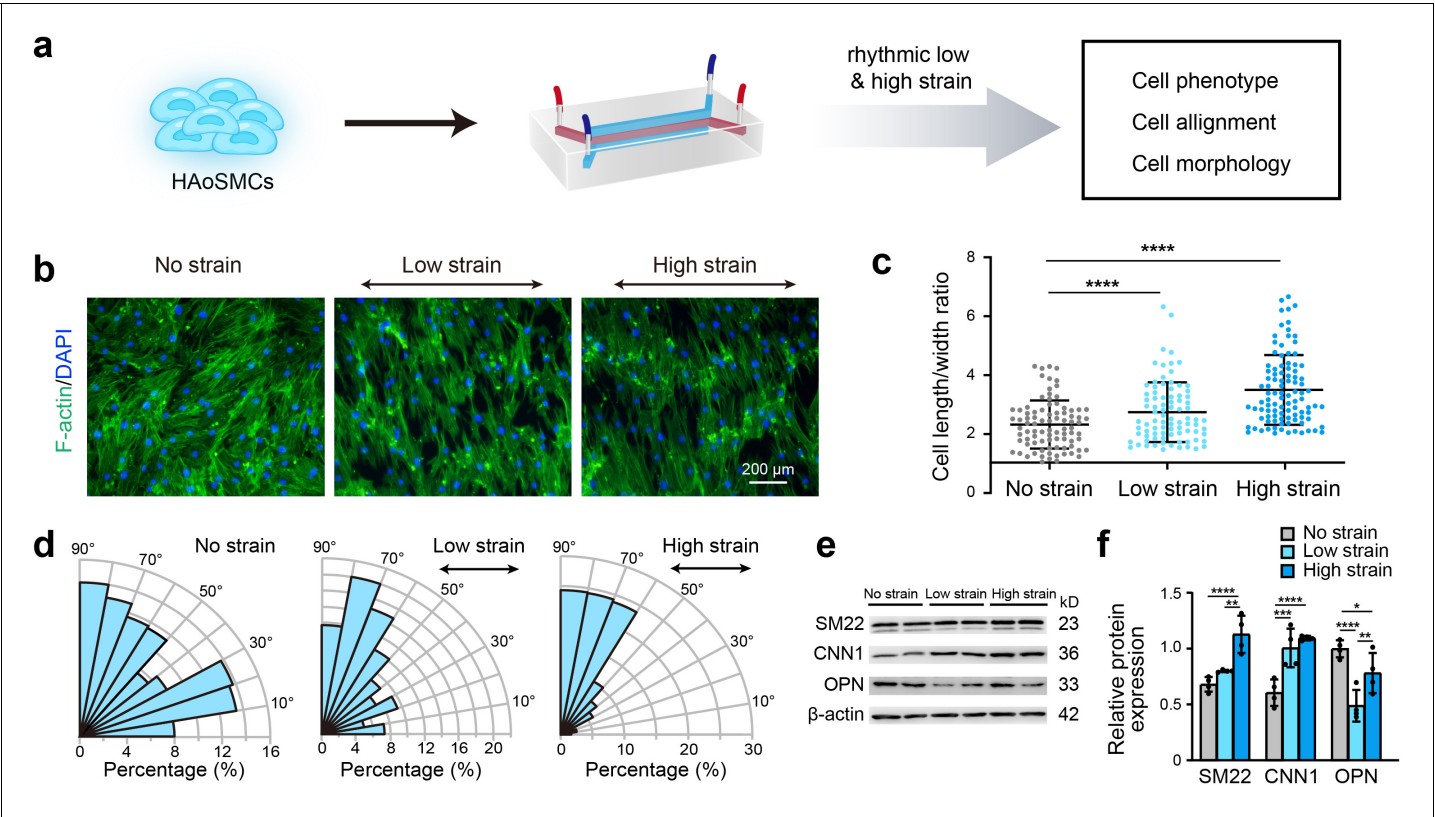

**Figure 3.** Effect of rhythmic strain on the cell morphology, alignment, and phenotype. (**a**) Schematic workflow of cell culture on the chip model. (**b**) Representative images of cytoskeletal F-actin staining of HAoSMCs exposed to low or high rhythmic strain for 24 hr. The scale bar represents 200 μm. (**c**) Length-to-width ratio of HAoSMCs after exposure to low or high rhythmic strain for 24 hr. (n = 3, cells were measured in three fields per sample,data from every single cell were plotted. ****p < 0.0001, one-way ANOVA followed by Tukey's post hoc test). (**d**) Alignments of HAoSMCs exposed to low or high rhythmic strain for 24 hr. (**e**) Representative images of the western blotting analyses of protein markers of the contractile phenotype (SM22 and CNN1) and synthetic phenotype (OPN) of HAoSMCs after exposure to low or high rhythmic strain for 24 hr. (**f**) Quantification of the total band densities for individual proteins normalized to the corresponding band of β-actin (n = 4, *p < 0.05, **p < 0.01, ***p < 0.001, ****p < 0.0001, one-way ANOVA followed by Tukey's post hoc test). All the data are expressed as the means ± SDs.

The online version of this article includes the following source data for figure 3:

**Source data 1.** The quantification data for cell length-to-width ratio.
**Source data 2.** The quantification data for orientation of the cells.
**Source data 3.** The original raw data for western blotting.

WT and NC groups (*Figure 4—figure supplement 1a*). Under static conditions, the mRNA expression of SM22 and CNN1 was upregulated and that of OPN was downregulated in the NOTCH1-KD group compared with the WT and NC groups (*Figure 4—figure supplement 1a*). The same tendency was found for the protein expression levels by western blotting analysis (*Figure 4—figure supplement 1b–c*).

NOTCH1-KD and WT HAoSMCs were cultured on aorta smooth muscle-on-a-chip models under rhythmic strain and static conditions to characterize the expression of SM22 and CNN1. As shown in *Figure 4b–c*, immunofluorescent staining images showed that SM22 and CNN1 were upregulated in the NOTCH-KD group compared with the WT group under static conditions. However, the opposite results were observed under rhythmic strain conditions: the expression of SM22 and CNN1 in NOTCH-KD HAoSMCs was lower than that in the WT group under rhythmic strain conditions. Western blotting analyses revealed similar alterations in the expression of SM22 and CNN1 (*Figure 4d–e*). In WT HAoSMCs, rhythmic strain induced the upregulation of SM22 and CNN1 expression compared with the levels observed under static conditions. However, a downregulation of SM22 and CNN1 expression was detected in NOTCH-KD HAoSMCs exposed to rhythmic strain. These results

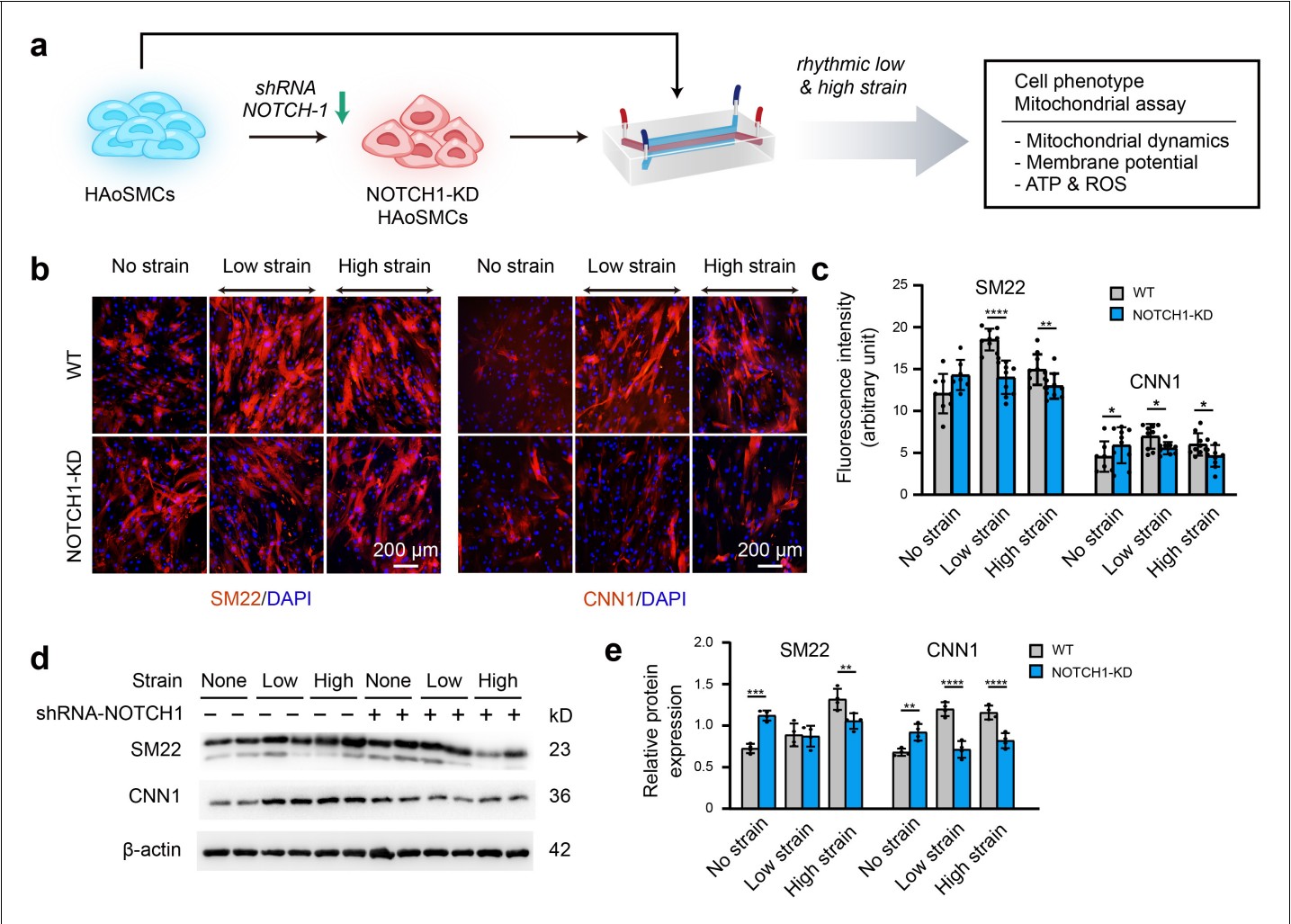

**Figure 4.** Phenotypic switching of NOTCH1-insufficient HAoSMCs under static and rhythmic strain conditions. (**a**) Schematic workflow of NOTCH1-KD HAoSMCs on the chip model. (**b**) Representative images of immunofluorescence staining of SM22 and CNN1 after exposure to rhythmic low or high strain for 24 hr. The scale bar represents 200 µm. (**c**) Intensity of immunofluorescence staining of SM22 and CNN1 (n=3, data from three independent biological replicates each with two to four technical replicates were plotted. *p < 0.05, **p < 0.01, ****p < 0.0001, two-way ANOVA followed by Tukey's post hoc test). (**d**) Representative images of the western blotting analyses of SM22 and CNN1 in the WT and NOTCH1-KD groups exposed to rhythmic low or high strains for 24 hr. (**e**) Quantification of the total band densities for individual proteins normalized to the corresponding band density of β-actin (n = 4, **p < 0.01, ***p < 0.001, ****p < 0.0001, two-way ANOVA followed by Tukey's post hoc test). All the data are expressed as the means ± SDs.

The online version of this article includes the following source data and figure supplement(s) for figure 4:

**Source data 1.** The data for intensity of immunofluorescence staining of SM22 and CNN1.

**Source data 2.** The original raw data for western blotting.

**Figure supplement 1.** Alteration of NOTCH1 signaling, phenotype and mitochondrial dynamics in HAoSMCs after NOTCH1 shRNA transfection.

**Figure supplement 1—source data 1.** The original raw data for western blotting.

suggested that rhythmic strain induced different effects between WT and NOTCH-KD HAoSMCs. In NOTCH-KD HAoSMCs, SM22 and CNN1 expression were higher under static conditions and lower under rhythmic strain than those in WT HAoSMCs.

Furthermore, we assessed the alterations in mitochondrial function and dynamics in NOTCH1-KD HAoSMCs. Specifically, western blotting was performed to detect the protein expression of MFN1 and MFN2 (mitochondrial fusion-related proteins) and DRP1 and MFF (mitochondrial fission-related proteins). No significant difference in MFN1, MFN2, DRP1, and MFF protein expression was found between the WT and NC groups under static conditions, which suggested that the expression of

these mitochondrial dynamics-related proteins was not affected by the lentivirus vector (*Figure 4—figure supplement 1d–e*). As shown in *Figure 5a–b*, the expression of MFN1 and MFN2 was down-regulated in NOTCH1-KD HAoSMCs under static conditions, and the expression of MFN2 presented a significant difference between the two groups. DRP1 and MFF expression was significantly increased in NOTCH1-KD HAoSMCs under static conditions. Rhythmic strain induced a decrease in MFN1 expression in the NOTCH1-KD group but an increase in the WT group. MFN2 expression was decreased in both the WT and NOTCH1-KD groups after exposure to rhythmic strain but was lower in the NOTCH1-KD group than in the WT group, particularly under rhythmic high strain conditions. Rhythmic strain induced an increase in DRP1 and MFF expression in both the WT and NOTCH1-KD groups, and DRP1 and MFF were highly expressed in NOTCH1-KD HAoSMCs under both rhythmic strain and static conditions (*Figure 5b*). The fragmented mitochondrial morphology observed in NOTHC1-KD HAoSMCs under rhythmic strain conditions indicated decreased mitochondrial fusion, increased mitochondrial fission, that is, impaired mitochondrial dynamics. As shown in *Figure 5c*, MitoTracker staining of the mitochondrial shape indicated elongated and interconnected mitochondrial networks in both WT and NOTCH1-KD groups under static conditions. Low or high strain induced alterations of the mitochondrial shape in both WT and NOTHC1-KD HAoSMCs, and these alterations included both a decrease in long rod-shaped elongated interconnected mitochondrial networks and a significant increase in spheroid-shaped fragmented mitochondrial morphology and were more frequent in NOTHC1-KD HAoSMCs under high strain conditions (*Figure 5c–d*).

Mitochondrial dynamics play an important role in the maintenance of normal mitochondrial function. As shown in *Figure 5e–f*, the fluorescence intensity of the tetramethylrhodamine methyl ester perchlorate (TMRM) staining, which reflects the mitochondrial membrane potential, was lower in NOTHC1-KD HAoSMCs than in the WT group under rhythmic strain. These results indicated loss of mitochondrial membrane potential in NOTCH1-KD HAoSMCs under rhythmic strain. The mitochondrial superoxide (MitoSOX) staining of NOTHC1-KD HAoSMCs was significantly higher than that of the WT group under rhythmic strain (*Figure 5g–h*), which indicated that ROS production was increased in NOTCH1-KD HAoSMCs under rhythmic strain conditions. As an energy source of cells synthesized by mitochondria, the ATP concentration was also evaluated. Rhythmic strain increased the ATP concentration in both WT and NOTHC1-KD HAoSMCs, but under rhythmic strain conditions, a lower ATP concentration was found in NOTCH1-KD HAoSMCs than in WT HAoSMCs (*Figure 5i*). No significant difference in the TMRM or MitoSOX fluorescence intensity or ATP concentration was found between WT and NOTCH1-KD HAoSMCs under static conditions, which indicated that NOTCH1 insufficiency did not affect mitochondrial function under static conditions. Taken together, these data indicated that NOTCH1 insufficiency could induce mitochondrial dysfunction in HAoSMCs by reducing mitochondrial fusion, inducing loss of mitochondrial membrane potential, increasing ROS production and generating insufficient ATP under rhythmic strain, and these effects are accompanied by an impaired contractile phenotype. These findings were consistent with previous studies showing that imbalanced mitochondrial dynamics could induce VSMC dedifferentiation into the synthetic phenotype (*Salabei and Hill, 2013*).

## Drugs rescued the impairment of mitochondrial dynamics in NOTCH1-insufficient HAoSMCs from BAV-TAA

To confirm whether the decreased contractile phenotype of NOTCH1-knockdown HAoSMCs can be rescued by inhibition of mitochondrial fission or activation of mitochondrial fusion, we evaluated the phenotypic alterations and mitochondrial dynamics of NOTCH1-insufficient HAoSMCs treated with a mitochondrial fission inhibitor (Mdivi-1) or mitochondrial fusion activators (leflunomide or teriflunomide) under rhythmic high-strain conditions. A schematic workflow of the drug screening experiments was shown in *Figure 6a*. Non-aneurysmal HAoSMCs with NOTCH1 knockdown and BAV-TAA HAoSMCs were used in drug testing experiments. The expression of MFN1 and/or MFN2 was reduced in NOTCH1-knockdown HAoSMCs of non-aneurysmal patient #1, #2, #3 and cells of BAV-TAA patient #2, #3 under rhythmic high-strain condition compared to static condition. Also, the cellular expression of SM22 and/or CNN1 was reduced in non-aneurysmal patient #1, #2 and BAV-TAA patient #1, #2, #3 under rhythmic high-strain condition compared to static condition. (*Figure 6—figure supplement 1*).

In the model using CRL1999 HAoSMCs (*Figure 6b*), all three drugs enhanced MFN1 and MFN2 expression to different extents, and the greatest increase in MFN1 expression was obtained with

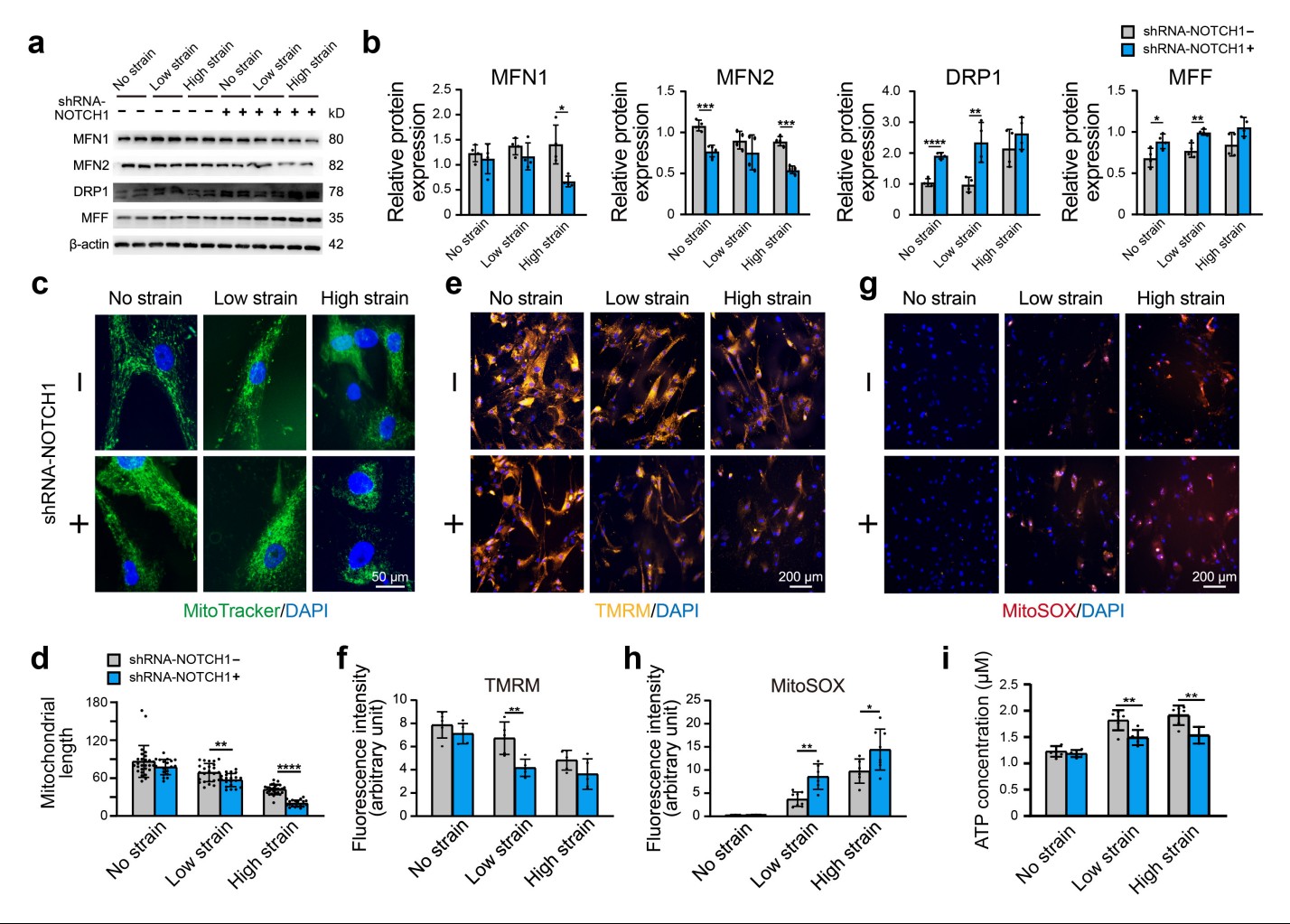

**Figure 5.** Effect of NOTCH1 insufficiency on mitochondrial dynamics and function in HAoSMCs. (a) Representative images of the western blotting analyses of MFN1, MFN2, DRP1 and MFF expression in the WT and NOTCH1-KD groups under rhythmic low or high strain or static conditions. (b) Quantification of the total band densities for four individual proteins normalized to the corresponding band density of β-actin (n = 4, *p < 0.05, **p < 0.01, ***p < 0.001, ****p < 0.0001, two-way ANOVA followed by Tukey's post hoc test). (c) MitoTracker staining images of the mitochondrial morphologies in the WT and NOTCH1-KD groups under rhythmic low or high strain or static conditions. The scale bar represents 50 μm. (d) Quantification of the mitochondria length (n = 3, **p < 0.01, ****p < 0.0001, two-way ANOVA followed by Tukey's post hoc test). (e) TMRM staining of the mitochondrial membrane potentials in the WT and NOTCH1-KD groups under rhythmic low or high strain or static conditions. The scale bar represents 100 μm. (f) Quantification of the relative TMRM fluorescence intensity (**p < 0.01, two-way ANOVA followed by Tukey's post hoc test). (g) MitoSOX staining of mitochondrial superoxide generation in the WT and NOTCH1-KD groups. The scale bar represents 100 μm. (h) Quantification of the relative MitoSOX fluorescence intensity (n=3, data from three independent biological replicates each with two to three technical replicates were plotted. *p < 0.05, **p < 0.01, two-way ANOVA followed by Tukey's post hoc test). (i) The ATP concentrations were measured using an ATP Determination Kit (n=3, data from three independent biological replicates each with two technical replicates were plotted. **p < 0.01, two-way ANOVA followed by Tukey's post hoc test). Quantitative measurements were calculated using ImageJ software. All the data are expressed as the means ± SDs. The online version of this article includes the following source data for figure 5:

**Source data 1.** The quantification data for mitochondria length.
**Source data 2.** The quantification data for relative TMRM fluorescence intensity.
**Source data 3.** The quantification data for relative MitoSOX fluorescence intensity.
**Source data 4.** The data for ATP concentrations.
**Source data 5.** The original raw data for western blotting.

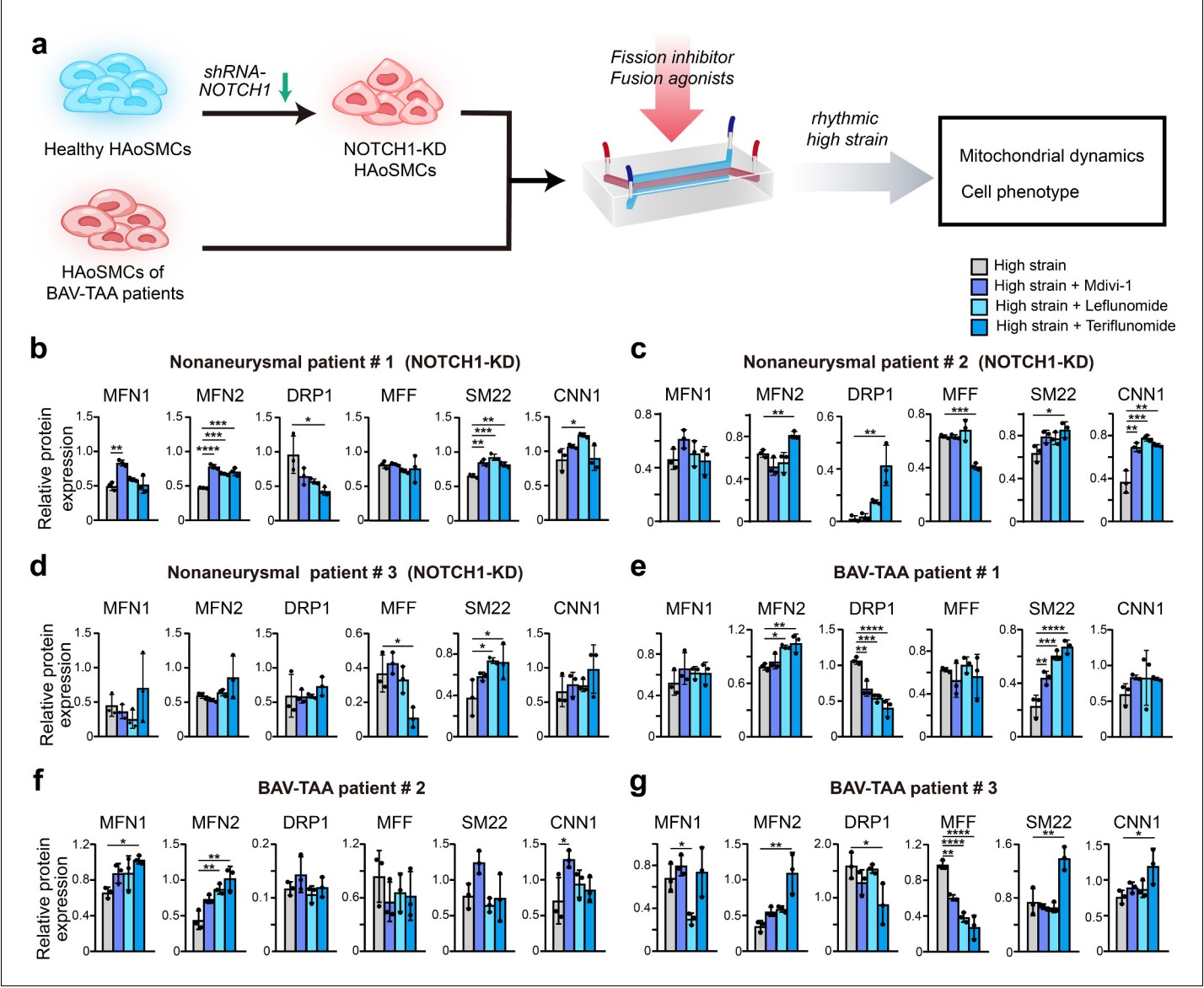

**Figure 6.** Screening of drugs that can rescue the cell phenotype and mitochondrial dynamics using the chip model. (a) Schematic workflow of the experimental design. After treatment with treated with Mdivi-1, leflunomide and teriflunomide on the chip models with (b) a NOTCH1-knockdown HAoSMC cell line (CRL1999), (c) NOTCH1-knockdown p-HAoSMCs isolated from non-diseased aortic tissues, (d) NOTCH1-knockdown p-HAoSMCs purchased from ATCC, and (e–g) p-HAoSMCs isolated from aortic tissues from three patients with BAV-TAA, quantification of the total band densities of the mitochondria-related proteins MFN1, MFN2, DRP-1, and MFF and the contractile phenotype proteins SM22 and CNN1 were normalized to the corresponding band densities of β-actin. All the data are expressed as the means ± SDs. n = 3, *p < 0.05, **p < 0.01, ***p < 0.001, ****p < 0.0001, one-way ANOVA followed by Tukey's post hoc test.

The online version of this article includes the following source data and figure supplement(s) for figure 6:

**Source data 1.** The original raw data for western blotting.

**Figure supplement 1.** Expressions of phenotype and mitochondrial dynamics related proteins in NOTCH1-knockdown HAoSMCs and p-HAoSMCs from BAV-TAA aortic tissues.

**Figure supplement 1—source data 1.** The original raw data for western blotting.

**Figure supplement 1—source data 2.** The original raw data for western blotting.

**Figure supplement 2.** Expressions of NOTCH1, phenotype and mitochondrial dynamics related proteins in p-HAoSMCs from BAV-TAA aortic tissues.

Mdivi-1. In addition, the three drugs all decreased DRP1 expression to different extents but did not affect MFF expression. The three drugs also increased SM22 and CNN1 expression to different extents, and the maximal expression of SM22 and CNN1 was obtained with leflunomide. In p-HAoSMCs isolated from patients in the non-diseased group (*Figure 6c*), the three drugs increased MFN1 expression, but no significant difference was found compared with the control group. Teriflunomide significantly increased MFN2 and DRP1 expression and significantly decreased MFF expression. Although the effects of these drugs on mitochondrial dynamics were not desirable, all three drugs increased the expression of SM22 and CNN1 to different extents. In p-HAoSMCs (ATCC) (*Figure 6d*), none of the three drugs exerted an obvious effect on MFN1, MFN2, and DRP1 expression, and teriflunomide significantly decreased MFF expression. All three drugs increased the expression of SM22 and CNN1 to different extents.

To further evaluate the potential of these drugs for clinical use in the treatment of NOTCH1-insufficient patients with BAV-TAA, we isolated p-HAoSMCs from the aortic tissues of three individual patients with BAV-TAA. The clinical characteristics of three patients with BAV-TAA are shown in *Supplementary file 1e*. Significantly lower NOTCH1 protein expression was found in p-HAoSMCs from BAV-TAA aortic tissues compared with p-HAoSMCs from non-diseased aortic tissue (*Figure 6—figure supplement 2*). The three drugs exerted different effects on mitochondrial dynamics-related proteins (*Figure 6e–g*). All three drugs enhanced MFN2 expression to different extents, and maximal MFN2 expression was obtained with the treatment of teriflunomide. The three drugs increased MFN1 expression and decreased DRP1 expression to different extents in two lines of BAV-TAA p-HAoSMCs, and the greatest decrease in DRP1 expression was obtained with teriflunomide. The three drugs decreased MFF expression to different extents in the two lines of BAV-TAA p-HAoSMCs and exerted different effects on the expression of the contractile proteins SM22 and CNN1, and teriflunomide significantly increased SM22 expression in the two lines of BAV-TAA p-HAoSMCs. All three drugs enhanced CNN1 expression to different extents, and Mdivi-1 or teriflunomide significantly increased CNN1 expression in one line of BAV-TAA p-HAoSMCs.

Different individual cases exhibited unevenness in drug response. Racial differences may lead to different cellular response to experimental stimuli or to drug treatment. HAoSMCs used in this study were from different human races. One primary HAoSMC line (ATCC, CRL1999) was isolated from a Caucasian donor and another one (ATCC, PCS-100–012) was isolated from an African American donor. Other four primary HAoSMCs were isolated from East Asian patients. Differences in patients' clinical characteristics, such as age, sex, and aortic diameter, may also have an impact on the experiment results. BAV aortopathy exclusively involves proximal aorta, including aortic root, ascending aorta and aortic arch, but spares the descending aorta and abdominal aorta. The majority of VSMCs at the proximal aorta and descending aorta arise from lineages of neural crest and paraxial mesoderm respectively. The heterogeneity of VSMCs could lead to section-specific microphysiology in the aortic wall and differences in the vulnerability of VSMCs to pathogenic stimuli. Thus, for the purpose of minimizing bias in drug response, HAoSMCs derived exclusively from the proximal aorta should be preferentially utilized in BAV aortopathy models. Overall, the data from BAV-TAA p-HAoSMCs indicated that teriflunomide exerted a more obvious effect on rescuing impaired mitochondrial dynamics and the expression of the contractile phenotype proteins SM22 and CNN1 under rhythmic high-strain conditions. These results indicated that leflunomide, Mdivi-1, and, in particular, teriflunomide, could serve as drug candidates for ameliorating the disease progression of BAV-TAA.

## Discussion

In this study, protein analyses of human aortic aneurysm tissues suggested the insufficient expression of NOTCH1 in BAV-TAA was associated with the impaired mitochondrial dynamics and OXPHOS. To verify it, we established a microfluidic model of aorta smooth muscle-on-a-chip, inspired by previous organ-on-chips models (*Ribas et al., 2017*; *Yasotharan et al., 2015*; *Huh et al., 2010*), enabling us to reproducibly generate the rhythmic tensile strain of the native human aortic wall. We showed that this model could apply a biomimetic rhythmic tensile strain with a confined amplitude/magnitude and rhythm to HAoSMCs. We also confirmed that impaired mitochondrial fusion contributed to the attenuated contractile phenotype observed in NOTCH1-KD HAoSMCs. We further identified that MFN1/2 agonists and DRP1 inhibitors were able to reverse the imbalance in mitochondrial dynamics and partly rescue the contractile phenotype in NOTCH1-insufficient HAoSMCs.

The majority of studies on the molecular mechanism and pharmacotherapy for TAA have been conducted using conventional cell culture models and animal models. The pathophysiological features of human aortic aneurysm can be induced in animal models by elastase, calcium chloride, angiotensin II or transgenesis, which partially enable investigation of the etiology, pathogenesis, and therapeutic targets of TAA at an early stage (*Eckhouse et al., 2013*; *Ikonomidis et al., 2003*; *Mao et al., 2015*). However, all these animal models cannot replicate the native characteristics of BAV-TAA. Moreover, the apparent species gap between preclinical animal experiments and clinical studies might also impede pharmaceutical discovery (*Lindeman and Matsumura, 2019*). In contrast, the shortcoming of the conventional cell culture model of aortic disease lies in its inability to recapitulate in vivo biomechanical stimuli. Advances in microfabrication and microfluidics have provided novel techniques for building organ-on-a-chip models that integrate rhythmic tensile strains into in vitro cell culture. Differently from conventional two dimensional (2D) and three dimensional (3D) cell culture methods, which are widely used in biological research, organ-on-a-chip models can replicate multicellular architectures, tissue-tissue interfaces, and biomechanical forces that exist in vivo, and precisely pattern cells and manipulate various mechanical and chemical parameters, such as flow rate, stretch, pressure, oxygen, and pH, providing controllable culture conditions not possible with conventional cultures (*Ahadian et al., 2018*). In the case of cardiovascular research, the functionally important cardiomyocytes, endothelial cells and smooth muscle, are constantly subjected to hemodynamic factors in vivo, including blood flow shear stress, rhythmic strain and fluid pressure, which cannot be simulated and given to the cells by conventional cell culture techniques. Organ-on-a-chip models can be broadly defined as any form of microfabricated cell culture device that models organ-specific biochemical and physical microenvironments (*Zhao et al., 2019*; *Park et al., 2019*) and represent an in vitro platform that is complementary to animal models.

The present aorta smooth muscle-on-a-chip model was designed based on the inspiration and reference of pioneering researchers' works, such as progeria-on-a-chip (*Ribas et al., 2017*), artery-on-a-chip (*Yasotharan et al., 2015*), and so on. Previous reported 'artery-on-a-chip' platforms are mainly focused on peripheral vessel, cerebral artery, or other arterial diseases. Yasotharan et al. reported an artery-on-a-chip platform that mimicked in vivo transmural pressure of an outer diameter of 120 µm olfactory artery segment, in which vascular tone and calcium dynamics were simultaneously assessed (*Poussin et al., 2020*). Poussin et al. have established a model of microvessels-on-a-chip under flow using primary human coronary arterial endothelial cells, to measure the adhesion of monocytes to the lumen of perfused microvessels (*Poussin et al., 2020*). Arterioles with a diameter of 15 µm, branches of the large artery, are mainly affected by fluid shear but receive little tensile tension. In contrast, aorta diameter is about 25–35 mm with thick three layers, in which smooth muscle cells are the main component in tunica media and experience high tensile strain. Aortopathies are the degenerative changes in the aortic wall inducing thinning or even rupture of the aorta. Thus, HAoSMCs in tunica media and the tensile strain they receive are essential components for the success of aorta smooth muscle-on-a-chip model.

By anatomizing the complexity of the biomechanical microenvironment, this study focused on the physiological rhythmic strain on HAoSMCs in the aortic tunica media, which plays a central role in the pathogenesis of aneurysms. In particular, the strain of the ascending aorta, which was defined as the maximal change in the ascending aorta diameter over a cardiac cycle, can vary diversely from a low value of 7.0 ± 2.5% to a high value of 21.5 ± 12.4% depending on age, sex and disease severity (*Akazawa et al., 2016*; *Bell et al., 2014*; *Morrison et al., 2009*). Under rhythmic strain, healthy HAoSMCs maintain a contractile phenotype, which allows regulation of the vascular tone. While, diseased HAoSMCs dedifferentiate into a synthetic phenotype under pathological conditions, and this dedifferentiation represents the initial step toward an aneurysm pathology (*Petsophonsakul et al., 2019*). The aorta smooth muscle-on-a-chip model developed in this study featured the controlled magnitude and rhythm of mechanical stimuli relevant to the human pathophysiological parameters of aorta biomechanics. In this study, HAoSMCs exhibited longer shapes in morphology, align unidirectionally, and present contractile phenotype on the on-chip model. In static conventional condition, HAoSMCs present more synthetic phenotype, which is oppositely different from the phenotype exist in native normal aortic wall. Most importantly, the on-chip model recapitulated the imbalanced mitochondrial dynamics which was accordant with analyses on tissues from human BAV-TAA and in vivo mouse model of abdominal aortic aneurysm (*Cooper et al., 2021*). In short, organ-on-a-chip models can mimic the biomechanical parameters, which are essential for aortopathy disease

development and more fully explore the pathophysiological changes of the cells and their real responsiveness to drugs, in a complementary manner with conventional cell culture and in vivo models. However, both of extracellular matrix and vascular cells, including epithelial cells, VSMCs, fibroblasts, macrophages, contribute to the pathological process of the aortic diseases. The present reductionist on-chip model merely represented HAoSMCs activities and did not recapitulated the entire aortic wall. Future works are expected to incorporate more components, especially the aortic extracellular matrix, into the on-chip model and explore ECM degradation and interactive molecular mechanisms involved in human aortopathies.

NOTCH1 has an important role in the maintenance of normal mitochondrial dynamics. Previous studies have reported that the expression of NOTCH1 signaling was attenuated in the aortic tissues of patients with BAVs (*Sciacca et al., 2013*). Additionally, Balistreri et al. found decreased mRNA levels of NOTCH1-4 in aortic aneurysm tissues and decreased protein levels of soluble NOTCH1 in plasma (*Balistreri et al., 2018*). Using clinical samples, our protein analysis also indicated that patients with BAV-TAA presented lower expression of NOTCH1 protein in ascending aortic aneurysms; and the NOTCH1 insufficiency was accompanied with attenuated mitochondrial fusion. This was consistent with the previous studies that reported NOTCH1 regulates mitochondria dynamics in cardiovascular cell differentiation and survival. Kasahara et al. found that mitochondrial fusion regulates the differentiation of myocardial cells through NOTCH1 signaling pathway (*Kasahara et al., 2013*). Here, we found that NOTCH1 insufficiency induced mitochondrial dysfunction in HAoSMCs by reducing mitochondrial fusion, along with loss of mitochondrial membrane potential, increasing ROS production and generating insufficient ATP under rhythmic strain. These results indicated that the relationship between impaired mitochondrial fusion and NOTCH1 deficiency in BAV-TAA. Consistently, we found decreased contractile phenotype in NOTCH1-deficient HAoSMCs under rhythmic tensile strain and in BAV-TAA aortic tissues, accompanied by lower NOTCH1 expression and impaired mitochondrial fusion.

NOTCH1 signaling is also involved in the regulation of HAoSMCs differentiation and the expression of contractile proteins. HAoSMCs are highly plastic cells and undergo reversible changes in phenotype in response to environmental stimuli (*Vásquez-Trincado et al., 2016*). In the aorta, HAoSMCs are considered to have the functional roles of both maintaining aortic tone in response to hemodynamic stimuli, and synthesizing and modeling the ECM (*Michel et al., 2018*). Most healthy HAoSMCs in the vascular wall in vivo exhibit a contractile phenotype, which allows them to maintain vascular tone (*Milewicz et al., 2008*). HAoSMCs contractile units associated proteins, such as SM22, CNN1, MYH11, and α-SMA, distribute the force on the aortic wall through regulation of extracellular matrix. Decreases in α-SMA, MYH11, SM22, and CNN1 attenuate HAoSMCs contractility unit formation and further disrupt force generation, promoting the development of aortic aneurysm or dissections (*Gillis et al., 2013*). Differentiated HAoSMCs have a contractile phenotype with little proliferation and little secretion of extracellular matrix. HAoSMCs transform from a contractile into a synthetic phenotype and induce loose of vascular tone, which is one of the major pathological process in TAA. Thus, the enhancement of HAoSMCs contractility is indicative of the effectiveness of pharmacotherapy for controlling aortic aneurysm (*Oller et al., 2021*). In this study, we found that NOTCH1 insufficiency in HAoSMCs downregulated the contractile phenotype proteins SM22 and CNN1 under rhythmic strain. Also, the expressions of SM22 and CNN1 were decreased in BAV-TAA aortic tissues compared to non-diseased aortic tissues. Noseda et al. reported that NOTCH1 activation was needed for expression of the contractile phenotype protein αSMA in VSMCs *via* the NOTCH/CSL axis (*Noseda et al., 2006*). High et al. reported that NOTCH1 promotes the differentiation of the cardiac neural crest into differentiated VSMCs and the expression of αSMA (*High et al., 2007*). However, the specific mechanism by which NOTCH1 leads to a reduced contractile phenotype has not been fully investigated. Based on results obtained from the aorta smooth muscle-on-a-chip model, it comes to the assumption that NOTCH1-insufficiency reduced the contractile phenotype through decreasing mitochondrial fusion.

Mitochondrial dynamics itself has been recognized as one of the critical factors that regulate the phenotype switching of vascular smooth muscle cells (VSMCs). Salabei and Hill reported that PDGF-BB induces the dedifferentiation of VSMCs into the synthetic phenotype through overactivation of mitochondrial fission (*Salabei and Hill, 2013*). Chen et al. found that the expression of MFN2 is markedly reduced in hyperproliferative VSMCs from spontaneously hypertensive rat arteries (*Chen et al., 2004*). Using the aorta smooth muscle-on-a-chip model, we demonstrated the

decreased mitochondrial fusion in NOTCH1-insufficient HAoSMCs, accompanied with decreased contractile phenotypes. Thus, mitochondrial dynamics regulated by NOTCH1 may have a role in maintains of contractile phenotype. In addition, we also found the contractile phenotype was upregulated in NOTCH1-insufficient HAoSMCs after treatment with mitochondrial fusion activators and mitochondrial fission inhibitors, which further verified the hypothesis that contractile phenotype of HAoSMCs was regulated by NOTCH1-mitochondrial fusion axis. In clinical samples, our protein analysis indicated that aortic tissues from patients with BAV-TAA presented the reduced expressions of mitochondrial fusion proteins and contractile phenotype proteins, compared with non-diseased aortic tissues. Further, the mass spectrometry-based proteomic analysis of aortic tissue samples indicated that mitochondrial dysfunction and OXPHOS pathways were significantly altered in BAV-TAA aortic tissues. Consistent with the previous single-cell transcriptome analysis, TAA aortic tissues showed decreased mitochondrial function; and this finding suggests that OXPHOS ATP production might be insufficient for VSMC contractile activities (*Li et al., 2020*). Therefore, maintaining normal mitochondrial dynamics *via* NOTCH1 could be one of the mechanisms through which NOTCH1 regulates HAoSMCs's contractility and maintains vascular wall tension.

The mitochondria morphology, that is fragmentation or elongation, which is controlled by precisely regulated mitochondrial fusion and fission, has been related to cardiovascular disorders, such as atherosclerosis and myocardial infarction (*Murphy et al., 2016*). Excessive mitochondrial fission induced a reduction in mitochondrial membrane potential and contractile phenotype of vascular VSMCs, and an increase in oxygen species production (*Lim et al., 2015*). These effects induced by mitochondrial fission were prevented by Mdivi-1, which is an inhibitor of mitochondrial fission related protein DRP1. Cooper et al. found that the upregulated DRP1 and mitochondrial fission in mouse abdominal aortic aneurysm, associated with impaired mitochondrial function and decreased contractility of mouse VSMCs (*Cooper et al., 2021*). The induction of the contractile to synthetic phenotype switch of VSMCs by platelet-derived growth factor, was also found to be associated with mitochondrial fragmentation/fission and attenuated MFN2 protein levels (*Salabei and Hill, 2013*). In this study, the excessive mitochondrial fragmentation of HAoSMCs implied diseased phenotype, while the restoration of mitochondrial homeostasis, i.e. the balance of fragmentation/fission and elongation/fusion, implied the rescue of HAoSMC contractility abnormality. However, the specific mechanism by which reduced mitochondrial fusion causes a reduction in the contractile phenotype requires further investigation. In addition, our proteomic analysis of tissue samples showed that several mitochondrial dysfunction-related pathways, namely, EIF2 signaling and the sirtuin signaling pathway, were also altered in BAV-TAA aortic tissue, which may provide research directions to investigate further mechanisms.

Lastly, we demonstrated that MFN1/2 agonists and DRP1 inhibitors reversed the imbalanced mitochondrial dynamics in BAV-TAA p-HAoSMCs with intrinsic NOTCH1 insufficiency or in NOTCH1-KD HAoSMCs on the aorta smooth muscle-on-a-chip model, which provides a prototype for an in vitro drug testing platform. Two FDA-approved drugs, leflunomide or teriflunomide, rescued the contractile phenotype of BAV-TAA p-HAoSMCs and showed their potential to ameliorate disease progression. Zorzano's group first found that dihydroorotate dehydrogenase inhibitors (leflunomide, teriflunomide, or brequinar sodium) could promote mitochondrial elongation through induction of the mitochondrial proteins MFN1 and MFN2 in HeLa cells (*Miret-Casals et al., 2018*). The experimental data obtained in this study encourage further studies on the application of leflunomide and teriflunomide in BAV-TAA.

This study has several limitations. Although the phenotypic reversal has observed after treatments of mitochondrial fission inhibitor and fusion agonists based on aorta smooth muscle-on-chip model, in vivo genetically modified mouse experiments are still needed to further confirm the therapeutic effects, that is whether these drugs can halt the progression of the disease. Also, genetically modified mouse aortic aneurysm model is required to further elucidate the genetic mechanism of NOTCH1—mitofusin axis in ascending aortic aneurysm. At the current stage, aorta smooth muscle-on-chip is still insufficient to alone predict clinical success, but it may be complementary with animal models in the sense that, together they are able to provide more comprehensive basis for preclinical assays with greater predictive power. Therefore, more verification tests, including in vitro on-chip tests and in vivo animal validations, are needed before translating the finding into prospective clinical tail.

In conclusion, we constructed an aorta smooth muscle-on-a-chip model, which could serve as a complementary tool to the current cell culture system and animal models. Using the aorta smooth muscle-on-a-chip model, we found that NOTCH1 insufficiency in HAoSMCs induced phenotypic switching from a contractile to a synthetic phenotype accompanied by an impairment of mitochondrial fusion, implying its potential role as a therapeutic target for BAV-TAA. At the current stage, in vitro microphysiological models and animal models are complementary to each other in the sense that they are able to provide more comprehensive basis for preclinical assays with greater predictive power.

## Acknowledgements

We Thank Drs. Fei Lan, Faxing Yu, Dan Ye, and Jiandong Ding at Fudan University for the helpful scientific discussions. Funding This work was supported by grants from the National Key R and D Program of China (2018YFC1005002), the National Natural Science Foundation of China (82070482, 81771971, 81772007, 21734003, and 51927805), the Shanghai Municipal Science and Technology Major Project (Grant No. 2017SHZDZX01), the Science and Technology Commission of Shanghai Municipality (17JC1400200, 20ZR1411700), and the Shanghai Municipal Education Commission (Innovation Program 2017-01-07-00-07-E00027).

## Additional information

### Funding

| Funder | Grant reference number | Author |
| --- | --- | --- |
| National Key Research and Development Program of China Stem Cell and Translational Research | 2018YFC1005002 | Weijia Zhang |
| National Natural Science Foundation of China | 82070482 | Weijia Zhang |
| National Natural Science Foundation of China | 81771971 | Kai Zhu |
| National Natural Science Foundation of China | 81772007 | Weijia Zhang |
| National Natural Science Foundation of China | 21734003 | Weijia Zhang |
| National Natural Science Foundation of China | 51927805 | Li Wang Kai Zhu |

The funders had no role in study design, data collection and interpretation, or the decision to submit the work for publication.

### Author contributions

Mieradilijiang Abudupataer, Conceptualization, Resources, Data curation, Formal analysis, Validation, Investigation, Visualization, Methodology, Writing - original draft, Writing - review and editing; Shichao Zhu, Validation, Investigation; Shiqiang Yan, Software, Formal analysis, Visualization, Methodology; Kehua Xu, Software, Formal analysis, Methodology; Jingjing Zhang, Formal analysis, Validation, Investigation; Shaman Luo, Yuyi Tang, Investigation, Methodology; Wenrui Ma, Resources, Formal analysis, Investigation; Md Fazle Alam, Writing - review and editing; Hui Huang, Formal analysis, Investigation; Nan Chen, Jun Li, Hao Lai, Resources; Li Wang, Validation, Writing - review and editing; Guoquan Yan, Software, Formal analysis; Chunsheng Wang, Conceptualization, Resources, Writing - review and editing; Kai Zhu, Conceptualization, Funding acquisition, Visualization, Writing - review and editing; Weijia Zhang, Conceptualization, Supervision, Funding acquisition, Visualization, Writing - original draft, Writing - review and editing

## Author ORCIDs

Mieradilijiang Abudupataer (iD) https://orcid.org/0000-0002-5421-9820
Shichao Zhu (iD) https://orcid.org/0000-0001-5057-9434
Kai Zhu (iD) https://orcid.org/0000-0002-2391-4193
Weijia Zhang (iD) https://orcid.org/0000-0001-6928-0416

## Ethics

Human subjects: Written informed consent was obtained from all patients before participation. Human aortic specimens were utilized under approvals of Zhongshan Hospital, Fudan University Ethics Committee (NO. B2020-158) in accordance with the Declaration of Helsinki.

## Decision letter and Author response

Decision letter https://doi.org/10.7554/eLife.69310.sa1
Author response https://doi.org/10.7554/eLife.69310.sa2

## Additional files

### Supplementary files

• Supplementary file 1. Supplementary information table 1 (a-e). (a) Clinical characteristics of the patients. (b) Primary antibodies used for western blotting and immunohistochemistry. (c) The enriched canonical pathways identified by ingenuity pathway analysis (IPA). (d) Differential protein expression of mitochondrial dysfunction and oxidative phosphorylation pathways between non-diseased and BAV-TAA tissues. (e) Clinical Characteristics of the p-HAoSMCs from BAV-TAA patients.

• Supplementary file 2. List of total differential protein expression between non-diseased and BAV-TAA tissues.

• Transparent reporting form

### Data availability

The mass spectrometry proteomics data, including raw data from the mass spectrometry runs, have been deposited to the ProteomeXchange Consortium via the PRIDE partner repository with the dataset identifier PXD026303. Theanalysis of the proteomics datasetare included in Figure 1.All other data supporting the findings of this study are included in the paper or supplementary files.

The following dataset was generated:

| Author(s) | Year | Dataset title | Dataset URL | Database and Identifier |
|---|---|---|---|---|
| Zhu K, Zhang W | 2021 | Aorta-on-a-chip reveals impaired mitochondrial dynamics as a therapeutic target for aortic aneurysm in bicuspid aortic valve disease | http://www.ebi.ac.uk/pride/archive/projects/PXD026303 | PRIDE, PXD026303 |

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
