## [Decision Letter]

**Acceptance summary:**

In this study, the authors have developed a microfluidics-based aorta-on-a-chip model and, using comparative proteomics analysis, identified potential mechanisms underlying the association between NOTCH1 signaling and BAV-TAA pathology, as well as evidence that impaired mitochondrial dynamics may be a therapeutic target. These studies are highly likely to drive in vivo experiments aimed at validation. The reviewers and editors feel this paper will be of great interest in the fields of organ-on-a-chip as well as bicuspid aortic valve disease and thoracic aortic aneurysm research.

**Decision letter after peer review:**

Thank you for submitting your article "Aorta-on-a-chip reveals impaired mitochondrial dynamics as a therapeutic target for aortic aneurysm in bicuspid aortic valve disease" for consideration by *eLife*. Your article has been reviewed by 3 peer reviewers, and the evaluation has been overseen by a Reviewing Editor and Balram Bhargava as the Senior Editor. The reviewers have opted to remain anonymous.

Essential revisions:

1. Aortic smooth muscle cells and its related tunica media play a central role in the development of aortopathy, but multiple other types of cells, including endothelial cells, fibroblasts, and macrophages, are also involved in the aortic wall and contribute to the pathological process. This work could inspire future work of integration of other cell types on the chip to study their network effect involved in the aortopathy, but the present study cannot represent the whole aortic wall on the chip. The authors should rephrase the statement of 'aorta-on-a-chip' in the entire manuscript since the current wording leaves the impression that they integrate entire aortic cells. The "aorta smooth muscle-on-a-chip" or a name would be more accurate. The title should be similarly clarified. Additionally, the limitations of this approach/model need to be discussed, for example the inability to recapitulate pathologic changes associated with disease such as ECM degradation, etc.

2. The authors used primary smooth muscle cells form patients with BAV-TAA. However, the population of VSMCs comprising the aortic microstructures is heterogeneous. In the ascending aorta and aortic arch, VSMCs are derived from the neural crest. VSMCs in the descending thoracic aorta are derived from the paraxial mesoderm, while neural crest and secondary heart field-derived VSMCs intermingle in the aortic root. The heterogeneity of VSMCs could lead to section-specific microphysiology in the aortic wall and differences in the vulnerability of VSMCs to pathogenic stimuli. Therefore, the authors should clarify which section of aorta they used and explain why they used that section. Furthermore, the authors should include a discussion of how heterogeneity of the primary cells can contribute to variability in outcomes in this model and how it should be appropriately dealt with.

3. This appears to be the first report of aortic function unit-on-a-chip to study aortopathy, the authors should clarify in more detail what discriminates this study from the other reported 'artery-on-a-chip' platforms (Artery-on-a-chip platform for automated, multimodal assessment of cerebral blood vessel structure and function, Lab Chip 2015;15(12):2660-9.doi: 10.1039/c5lc00021a) and where the novelty is. The reviewers were all excited by this work, but the novelty should not be overstated and prior studies using similar models should be given due credit.

4. In Figure 6, the patients-derived cells were used on the chip. The effect of drug testing on aorta-on-a-chip model was desirable, even though the results did not exhibit all the positive responses. Please explain the potential reasons for the unevenness. Additionally, Figure 6 needs to include the control condition and statistical comparisons between this and the high strain condition.

5. A discussion regarding device design and optimization should be included. Did the authors try other designs? For instance, the stretching chip, demonstrated by Ingber's group, incorporates two lateral chambers for manipulating cyclic stretching forces on cells. Why was the chosen design selected, and what parameters are most critical to its success?

6. Details on chip fabrication must be included for the methodology to be adequate for publication. Detailed description on "commercialized highly flexible polydimethylsiloxane (PDMS) membranes", which is a key component of the chip, is needed. Definitions and descriptions of the ascending aorta, common value ranges for physiologic parameters, how they correlate with the vacuum pressure, etc, need to be discussed in the 'construction of aorta on a chip model' section rather than (or in addition to) the discussion.

7. In Figure 2—figure supplement 1 figure caption (b) Young's modulus of the PDMS membrane, the caption doesn't match the image. The image doesn't show the Young's modulus directly. What was the calculated value of Young's modulus? It seems the measured value was different from the commercially labeled value, 1.7 MPa, Line 548, Page 15. How did authors measure it? This is missing in the Materials and methods Part. Authors should describe measurement protocol in Methods and calculated value in Results. Figure 2—figure supplement 1 figure (b) should present a scatter plot instead of a line chart.

8. In Figure 2, the authors only presented the video snapshots. The authors should provide movie clips, or GIF animation images, for the purpose of repeating this experiment and evaluating results by readers.

9. While beyond the scope of this study, the on-a-chip model is clearly insufficient to alone predict clinical success, and the authors need to discuss the importance of following up on these studies using in vivo validation. This reductionist on-a-chip model is viewed as powerful for drug discovery, but not sufficient to motivate clinical trials without in vivo validation. Please add some discussion regarding these points.

10. Please specify the z-scores of individual pathways in Figure 1F, and the scale of the horizontal axis in Figure 1H (log2 or log10?).

11. Please add any discussion regarding implications of enhanced cell contractility and changes to mitochondria morphology.

12. Please discuss in greater detail the benefits of the on-a-chip model compared to conventional cell culture.

---

## [Author Response]

Essential revisions:1. Aortic smooth muscle cells and its related tunica media play a central role in the development of aortopathy, but multiple other types of cells, including endothelial cells, fibroblasts, and macrophages, are also involved in the aortic wall and contribute to the pathological process. This work could inspire future work of integration of other cell types on the chip to study their network effect involved in the aortopathy, but the present study cannot represent the whole aortic wall on the chip. The authors should rephrase the statement of 'aorta-on-a-chip' in the entire manuscript since the current wording leaves the impression that they integrate entire aortic cells. The "aorta smooth muscle-on-a-chip" or a name would be more accurate. The title should be similarly clarified. Additionally, the limitations of this approach/model need to be discussed, for example the inability to recapitulate pathologic changes associated with disease such as ECM degradation, etc.

To clarify it, we rephrased the statement of “aorta-on-a-chip” into “aorta smooth muscle-on-a-chip” in the revised manuscript. Also, we added limitation discussion, as follows.

Page 22, Line 755: “Both of extracellular matrix and vascular cells, including epithelial cells, VSMCs, fibroblasts, macrophages, contribute to the pathological process of the aortic diseases. The present reductionist on-chip model merely represented HAoSMCs activities and did not recapitulate the entire aortic wall. Future works are expected to incorporate more components, especially the aortic extracellular matrix, into the on-chip model and explore ECM degradation and interactive molecular mechanisms involved in human aortopathies.”

2. The authors used primary smooth muscle cells form patients with BAV-TAA. However, the population of VSMCs comprising the aortic microstructures is heterogeneous. In the ascending aorta and aortic arch, VSMCs are derived from the neural crest. VSMCs in the descending thoracic aorta are derived from the paraxial mesoderm, while neural crest and secondary heart field-derived VSMCs intermingle in the aortic root. The heterogeneity of VSMCs could lead to section-specific microphysiology in the aortic wall and differences in the vulnerability of VSMCs to pathogenic stimuli. Therefore, the authors should clarify which section of aorta they used and explain why they used that section. Furthermore, the authors should include a discussion of how heterogeneity of the primary cells can contribute to variability in outcomes in this model and how it should be appropriately dealt with.

(1) We harvested the primary cells from ascending aortic aneurysm with purpose. To clarify it, we added this information in the Methods section.

(2) We discuss how heterogeneity of the primary cells can contribute to variability and how to deal with it in the Discussion section.

Page 19, Line 652: “BAV aortopathy exclusively involves proximal aorta, including aortic root, ascending aorta and aortic arch, but spares the descending aorta and abdominal aorta. The majority of VSMCs at the proximal aorta and descending aorta arise from lineages of neural crest and paraxial mesoderm respectively. The heterogeneity of VSMCs could lead to section-specific microphysiology in the aortic wall and differences in the vulnerability of VSMCs to pathogenic stimuli. Thus, for the purpose of minimizing bias in drug response, HAoSMCs derived exclusively from the proximal aorta should be preferentially utilized in BAV aortopathy models.”

3. This appears to be the first report of aortic function unit-on-a-chip to study aortopathy, the authors should clarify in more detail what discriminates this study from the other reported 'artery-on-a-chip' platforms (Artery-on-a-chip platform for automated, multimodal assessment of cerebral blood vessel structure and function, Lab Chip 2015;15(12):2660-9.doi: 10.1039/c5lc00021a) and where the novelty is. The reviewers were all excited by this work, but the novelty should not be overstated and prior studies using similar models should be given due credit.

We revised a few sentences to avoid overstating the novelty and summarized artery-on-chip models that have been previous reported.

Page 21, Line 686“To verify it, we established a microfluidic model of aorta smooth muscle-on-a-chip, inspired by previous organ-on-chips models [1,2,3], enabling us to reproducibly generate the rhythmic tensile strain of the native human aortic wall.”

Page 21, Line 719“The present aorta smooth muscle-on-a-chip model was designed based on the inspiration and reference of pioneering researchers' works, such as progeria-on-a-chip [1], artery-on-a-chip [2], and so on. Previous reported 'artery-on-a-chip' platforms are mainly focused on peripheral vessel, cerebral artery, or other arterial diseases. Yasotharan et al. reported an artery-on-a-chip platform that mimicked in vivo transmural pressure of an outer diameter of 120 μm olfactory artery segment, in which vascular tone and calcium dynamics were simultaneously assessed [3]. Poussin et al. have established a model of microvessels-on-a-chip under flow using primary human coronary arterial endothelial cells, to measure the adhesion of monocytes to the lumen of perfused microvessels [3]. Arterioles with a diameter of 15 μm, branches of the large artery, are mainly affected by fluid shear but receive little tensile tension. In contrast, aorta diameter is about 25-35 mm with thick three layers, in which smooth muscle cells are the main component in tunica media and experience high tensile strain. Aortopathies are the degenerative changes in the aortic wall inducing thinning or even rupture of the aorta. Thus, HAoSMCs in tunica media and the tensile strain they receive are essential components for the success of aorta smooth muscle-on-a-chip model. “

References

1. Ribas J, Zhang YS, Pitrez PR, Leijten J, Miscuglio M, Rouwkema J, Dokmeci MR, Nissan X, Ferreira L, Khademhosseini A. Biomechanical Strain Exacerbates Inflammation on a Progeria-on-a-Chip Model. Small. 2017,15, 10-35

2. Yasotharan S, Pinto S, Sled JG, Bolz SS, Günther A. Artery-on-a-chip platform for automated, multimodal assessment of cerebral blood vessel structure and function. Lab Chip. 2015,12, 2660-2669.

3. Poussin C, Kramer B, Lanz HL, Van den Heuvel A, Laurent A, Olivier T, Vermeer M, Peric D, Baumer K, Dulize R, Guedj E, Ivanov NV, Peitsch MC, Hoeng J, Joore J. 3D human microvessel-on-a-chip model for studying monocyte-to-endothelium adhesion under flow – application in systems toxicology. ALTEX. 2020, 1, 47-63.

4. In Figure 6, the patients-derived cells were used on the chip. The effect of drug testing on aorta-on-a-chip model was desirable, even though the results did not exhibit all the positive responses. Please explain the potential reasons for the unevenness. Additionally, Figure 6 needs to include the control condition and statistical comparisons between this and the high strain condition.

(1) This unevenness may arise from individual differences and genetic background. We discussed the potential reasons for the unevenness, as follows.

Page 19, Line 646: “Different individual cases exhibited unevenness in drug response. Racial differences may lead to different cellular response to experimental stimuli or to drug treatment. HAoSMCs used in this study were from different human races. One primary HAoSMC line (ATCC CRL1999) was isolated from a Caucasian donor and another one (ATCC, PCS^-^100-012) was isolated from an African American donor. Other four primary HAoSMCs were isolated from East Asian patients. Differences in patients’ clinical characteristics, such as age, sex and aortic diameter, may also have an impact on the experiment results.”

(2) Control condition and statistical comparisons with high strain condition were added in the in Figure 6, as Figure 6 — figure supplement 2.

In the first submission, we conducted experiments repeats of control condition. But due to the limited sample loading wells on one SDS-PAGE gel, we only performed 4 groups (high strain and 3 drugs treatment) western blot analysis. We re-performed 3 repeated controls and 3 repeated high strain condition together and added statistical comparison in the revised manuscript.

We also added description in Results section.

Page 19, Line 612: “The expression of MFN1 and/or MFN2 was reduced in NOTCH1-knockdown HAoSMCs of non-aneurysmal patient #1, #2, #3 and cells of BAV-TAA patient #2, #3 under rhythmic high-strain condition compared to static condition. Also, the cellular expression of SM22 and/or CNN1 was reduced in non-aneurysmal patient #1, #2 and BAV-TAA patient #1, #2, #3 under rhythmic high-strain condition compared to static condition *(Figure 6—figure supplement 1).*”

5. A discussion regarding device design and optimization should be included. Did the authors try other designs? For instance, the stretching chip, demonstrated by Ingber's group, incorporates two lateral chambers for manipulating cyclic stretching forces on cells. Why was the chosen design selected, and what parameters are most critical to its success?

The cyclic stretching lung-on-chip with lateral vacuum chambers is a multifunctional design. But we did not use it and designed a chip with up and down vacuum chambers for the main reason of its easier fabrication. We tested three prototypes with varying culturing channels, i.e. 2, or 4, or 6 mm in width. Finally, we opted for the current size mainly because we aimed to harvest enough cells for multiple protein analysis replication. The miniaturized chip designs, though capable of culturing a small number of cells for PCR or immunostaining analysis, cannot support enough cells for multiple western blotting experiments which is viewed as one of the gold standards in molecular biology.

We also provided three “gif” format animations about the working status of three prototypes as supplementary data and added the above discussion in the revised manuscript.

Page 13, Line 419: “We tested three prototypes with varying culturing channels, i.e. 2, or 4, or 6 mm in width (Figure 2b-c and Figure 2—figure supplement 4). Finally, we opted for the largest size to harvest enough cells for protein analysis replication.”

6. Details on chip fabrication must be included for the methodology to be adequate for publication. Detailed description on "commercialized highly flexible polydimethylsiloxane (PDMS) membranes", which is a key component of the chip, is needed. Definitions and descriptions of the ascending aorta, common value ranges for physiologic parameters, how they correlate with the vacuum pressure, etc, need to be discussed in the 'construction of aorta on a chip model' section rather than (or in addition to) the discussion.

(1) Description on PDMS membrane.

This commercialized polydimethylsiloxane (PDMS) membrane is a company’s patented product. This uniform PDMS membrane, featured by outstanding strength and elasticity, is the core component to ensure the robustness and repeatability. We provided more mechanic tests as follows.

In the revised manuscript, we provide Figure 2— figure supplement 3, and detailed parameters of the commercial PDMS membrane in Materials and methods.

Page 6, Line 107: “*Detailed parameters of the commercialized PDMS membrane were provided as follows：thickness of 200 ± 2 μm, shore A hardness of 50, Yang's elastic modulus of 1.7 MPa, tensile strength of 4 MPa, tear strength of 7 KN/m and light transmittance of 93%”*

(2) The 'construction of aorta on a chip model' section.

In the revised manuscript, we added the related definitions and descriptions in the Results part.

Page 13, Line 390: “Ascending aorta is the first section of the aorta, which starts from the left ventricle of the heart and extends to the aortic arch. It is connected to the left ventricular outflow track and is the part that pumps oxygenated blood to the body's tissues and organs. Clinical studies have shown that the circumferential strains of the aortic wall range from low values of 7.0 ± 2.5% to high values of 21.5 ± 12.4%. We captured the real-time deformations of the PDMS membranes from a cross-sectional view of the microfluidic model, with vacuum pressures of 0 kPa, 10 kPa, and 15 kPa, and measured the strain magnitude of the PDMS membrane. “

7. In Figure 2—figure supplement 1 figure caption (b) Young's modulus of the PDMS membrane, the caption doesn't match the image. The image doesn't show the Young's modulus directly. What was the calculated value of Young's modulus? It seems the measured value was different from the commercially labeled value, 1.7 MPa, Line 548, Page 15. How did authors measure it? This is missing in the Materials and methods Part. Authors should describe measurement protocol in Methods and calculated value in Results. Figure 2—figure supplement 1 figure (b) should present a scatter plot instead of a line chart.

We revised the figure caption as “(b) Tensile stress–strain responses of PDMS membrane” and re-drew the scatter plots with the uploaded source data, in the revised manuscript. The measured Young's elastic modulus values were 1.71 MPa within 25% tensile strain and 1.67 MPa within 500% tensile strain. Overall, it is in agreement with commercially labeled value. We added the calculated Young's elastic modulus values and measurement methods in the revised manuscript.

Page 6, Line 110: “*The Young's modulus values were obtained by the following experiments. The tensile stress–strain responses was measured using a tensile testing machine (Instron). Prior to the measurement, the PDMS membrane was cut into a piece of 3 cm length and 3 mm width membrane. The membrane was fixed to the testing machine with a fixture. The sample was automatically stretched in a gradient within the proportional limit. The Young's modulus of the PDMS membrane was calculated by the slope value of the tensile stress–strain curve.*”

8. In Figure 2, the authors only presented the video snapshots. The authors should provide movie clips, or GIF animation images, for the purpose of repeating this experiment and evaluating results by readers.

We added a “gif” format animation (gif image) as a supplementary data to Figure 2, as Figure 2—figure supplement 4.

9. While beyond the scope of this study, the on-a-chip model is clearly insufficient to alone predict clinical success, and the authors need to discuss the importance of following up on these studies using in vivo validation. This reductionist on-a-chip model is viewed as powerful for drug discovery, but not sufficient to motivate clinical trials without in vivo validation. Please add some discussion regarding these points.

We discussed limitation in revised manuscript, as follows.

Page 24, Line 857: “This study has several limitations. Although the phenotypic reversal has observed after treatments of mitochondrial fission inhibitor and fusion agonists based on aorta smooth muscle-on-chip model, in vivo genetically modified mouse experiments are still needed to further confirm the therapeutic effects, i.e., whether these drugs can effectively halt the progression of the disease. Also, genetically modified mouse aortic aneurysm model is required to further elucidate the genetic mechanism of NOTCH1—mitofusin axis in ascending aortic aneurysm. At the current stage, aorta-on-chip is still insufficient to alone predict clinical success, but it may be complementary with animal models in the sense that, together they are able to provide more comprehensive basis for preclinical assays with greater predictive power. Therefore, more verification tests, including in vitro on-chip tests and in vivo animal validations, are needed before translating the finding into prospective clinical tail.”

10. Please specify the z-scores of individual pathways in Figure 1F, and the scale of the horizontal axis in Figure 1H (log2 or log10?).

We added z-scores in the revised Figure 1F and specified Log2 in Figure 1H. The z-scores have also been provided in the source data associated with Figure 1. Z-score N/A indicates pathways for which no prediction can be made due to insufficient evidence in the Knowledge Base for confident activity predictions across datasets, as explained by the introduction of QIAGEN Ingenuity Pathway Analysis software.

We also revised the results description.

Page 11, Line 347: “Among these pathways, Z-score of oxidative phosphorylation pathway was -2.333, indicating significantly inhibited. Z-score of mitochondria dysfunction was not applicable due to insufficient evidence in the knowledge base for confident activity predictions across datasets.”

11. Please add any discussion regarding implications of enhanced cell contractility and changes to mitochondria morphology.

Implications of enhanced cell contractility and changes to mitochondria morphology were added to the discussion part respectively, as follows:

(1) Implications of enhanced cell contractility.

Page 22 Line 782: “In the aorta, HAoSMCs are considered to have the functional roles of both maintaining aortic tone in response to hemodynamic stimuli, and synthesizing and modelling the ECM [1]. Most healthy HAoSMCs in the vascular wall in vivo exhibit a contractile phenotype, which allows them to maintain vascular tone [2]. HAoSMCs contractile units associated proteins, such as SM22, CNN1, MYH11 and α-SMA, distribute the force on the aortic wall through regulation of extracellular matrix. Decreases in α-SMA, MYH11, SM22, and CNN1 attenuate HAoSMCs contractility unit formation and further disrupt force generation, promoting the development of aortic aneurysm or dissections [3]. Thus, the enhancement of HAoSMCs contractility is one of indicators of the effectiveness of pharmacotherapy for controlling aortic aneurysm [4].”

References

1. Michel JB, Jondeau G, Milewicz DM. From genetics to response to injury: vascular smooth muscle cells in aneurysms and dissections of the ascending aorta. Cardiovasc Res. 2018, 114, 578-589.

2. Milewicz, D. M. et al., Genetic basis of thoracic aortic aneurysms and dissections: focus on smooth muscle cell contractile dysfunction. Annu Rev Genomics Hum Genet. 2008, 9, 283–302.

3. Gillis E, Van Laer L, Loeys BL. Genetics of thoracic aortic aneurysm: at the crossroad of transforming growth factor-β signaling and vascular smooth muscle cell contractility. Circ Res. 2013 113, 327-340.

4. Oller J, Gabandé-Rodríguez E, Ruiz-Rodríguez MJ, et al., Extracellular Tuning of Mitochondrial Respiration Leads to Aortic Aneurysm. Circulation. 2021, 143, 2091-2109.

(2) Implications of mitochondria morphology changes.

Page 23 Line 828: “The mitochondria morphology, i.e., fragmentation or elongation, which is controlled by precisely regulated mitochondrial fusion and fission, has been related to cardiovascular disorders, such as atherosclerosis and myocardial infarction [1]. Excessive mitochondrial fission induced a reduction in mitochondrial membrane potential and contractile phenotype of vascular VSMCs, and an increase in oxygen species production [2]. These effects induced by mitochondrial fission were prevented by Mdivi-1, which is an inhibitor of mitochondrial fission related protein DRP1. Cooper et al. found that the upregulated DRP1 and mitochondrial fission in mouse abdominal aortic aneurysm, associated with impaired mitochondrial function and decreased contractility of mouse VSMCs [3]. The induction of the contractile to synthetic phenotype switch of VSMCs by platelet-derived growth factor, was also found to be associated with mitochondrial fragmentation/fission and attenuated MFN2 protein levels [4]. In this study, the excessive mitochondrial fragmentation of HAoSMCs implied diseased phenotype, while the restoration of mitochondrial homeostasis, i.e., the balance of fragmentation/fission and elongation/fusion, implied the rescue of HAoSMC contractility abnormality.”

References

1. Murphy E, Ardehali H, Balaban RS, DiLisa F, Dorn GW II, Kitsis RN, Otsu K, Ping P, Rizzuto R, Sack MN, Wallace D, Youle RJ. Mitochondrial function, biology, and role in disease: a scientific statement from the American Heart Association. Circ Res 2016,118,1960–1991.

2. Lim S, Lee SY, Seo HH, Ham O, Lee C, Park JH, Lee J, Seung M, Yun I, Han SM, Lee S, Choi E, Hwang KC. Regulation of mitochondrial morphology by positive feedback interaction between PKCdelta and Drp1 in vascular smooth muscle cell. J Cell Biochem 2015,116, 648–660.

3. Cooper HA, Cicalese S, Preston KJ, et al., Targeting mitochondrial fission as a potential therapeutic for abdominal aortic aneurysm. Cardiovasc Res. 2021, 22, 971-982.

4. Salabei JK and Hill BG. Mitochondrial fission induced by platelet-derived growth factor regulates vascular smooth muscle cell bioenergetics and cell proliferation. Redox Biol 2013, 1, 542–551.

12. Please discuss in greater detail the benefits of the on-a-chip model compared to conventional cell culture.

The benefits of the on-a-chip model compared to conventional cell culture were extended in the revised manuscript, as follows.

Page 21, Line 706: “Differently from conventional two dimensional (2D) and three dimensional (3D) cell culture methods which are widely used in biological research, organ-on-a-chip models can replicate multicellular architectures, tissue-tissue interfaces, and biomechanical forces that exist in vivo, and precisely pattern cells and manipulate various mechanical and chemical parameters, such as flow rate, stretch, pressure, oxygen, and pH, providing controllable culture conditions not possible with conventional cultures [1]. In the case of cardiovascular research, the functionally important cardiomyocytes, endothelial cells and smooth muscle, are constantly subjected to hemodynamic factors in vivo, including blood flow shear stress, rhythmic strain and fluid pressure, which cannot be simulated and given to the cells by conventional cell culture techniques.”

Page 22, Line 746: “In this study, HAoSMCs exhibited longer shapes in morphology, align unidirectionally, and present contractile phenotype on the on-chip model. In static conventional condition, HAoSMCs present more synthetic phenotype, which is oppositely different from the phenotype exist in native normal aortic wall. Most importantly, the on-chip model recapitulated the imbalanced mitochondrial dynamics which was accordant with analyses on tissues from human BAV-TAA and in vivo mouse model of abdominal aortic aneurysm [2]. In short, organ-on-a-chip models can mimic the biomechanical parameters, which are essential for aortopathy development and more fully explore the pathophysiological changes of the cells and their real responsiveness to drugs, in a complementary manner with conventional cell culture and in vivo models.”

References

1. Ahadian S, Civitarese R, Bannerman D, Mohammadi MH, Lu R, Wang E, Davenport-Huyer L, Lai B, Zhang B, Zhao Y, Mandla S, Korolj A, Radisic M. Organ-On-A-Chip Platforms: A Convergence of Advanced Materials, Cells, and Microscale Technologies. Adv Healthc Mater, 2018, 2, 1-53.

2. Cooper HA, Cicalese S, Preston KJ, et al., Targeting mitochondrial fission as a potential therapeutic for abdominal aortic aneurysm. Cardiovasc Res. 2021, 22, 971-982.